# Expression of a constitutively active human *STING* mutant in hematopoietic cells produces an *Ifnar1*-dependent vasculopathy in mice

Gary R Martin[1,2] , Kimiora Henare[1,2,6] , Carolina Salazar[1,2], Teresa Scheidl-Yee[1,2], Laura J Eggen[1,2], Pankaj P Tailor[1,2], Jung Hwan Kim[4,5], John Podstawka[5], Marvin J Fritzler[1,2], Margaret M Kelly[3,5], Bryan G Yipp[4,5], Frank R Jirik[1,2]

**STING-associated vasculopathy with onset in infancy (SAVI) is an autoinflammatory disorder characterized by blood vessel occlusions, acral necrosis, myositis, rashes, and pulmonary inflammation that are the result of activating mutations in the STimulator of Interferon Genes (STING). We generated a transgenic line that recapitulates many of the phenotypic aspects of SAVI by targeting the expression of the human STING-N154S–mutant protein to the murine hematopoietic compartment. *hSTING-N154S* mice demonstrated failure to gain weight, lymphopenia, progressive paw swelling accompanied by inflammatory infiltrates, severe myositis, and ear and tail necrosis. However, no significant lung inflammation was observed. X-ray microscopy imaging revealed vasculopathy characterized by arteriole occlusions and venous thromboses. Type I interferons and proinflammatory mediators were elevated in *hSTING-N154S* sera. Importantly, the phenotype was prevented in *hSTING-N154S* mice lacking the type I interferon receptor gene (*Ifnar1*). This model, based on a mutant human STING protein, may shed light on the pathophysiological mechanisms operative in SAVI.**

## Introduction

As an important component of a sensing mechanism for cytosolic dsDNA derived from viruses, bacteria, or the host, the STING protein has the ability to trigger potent type I interferon responses (Ishikawa et al, 2009; Abe et al, 2013; Gao et al, 2013; Xiao & Fitzgerald, 2013). However, de novo activating mutations in the STING molecule have also been identified as being responsible for a monogenic autoinflammatory syndrome (Jeremiah et al, 2014; Chia et al, 2016; Fremond et al, 2016; Picard et al, 2016; Konig et al, 2017) known as

SAVI (Liu et al, 2014). This autosomal dominant genetic disease has been attributed to a number of distinct gain-of-function *STING* mutations (also known as *TMEM173*) leading to the constitutive activation of the STING protein (Liu et al, 2014). The SAVI phenotype is characterized by blood vessel inflammation and damage, development of inflammatory skin lesions, losses of ear and nasal cartilages, as well as ulceration and necrosis of digits that often require amputation (Jeremiah et al, 2014; Liu et al, 2014; Chia et al, 2016). Additional features can include a lupus-like syndrome (Konig et al, 2017), arthralgias, myositis (Liu et al, 2014; Fremond et al, 2016), and potentially fatal lung disease (Liu et al, 2014; Picard et al, 2016).

Laboratory features of SAVI can include increased levels of inflammatory markers such as C-reactive protein and the erythrocyte sedimentation rate (Liu et al, 2014; Munoz et al, 2015), anemia, lymphocytopenia, thrombocytosis, hyper-γ-globulinemia, evidence of immune complex deposition, and the presence of antinuclear antibodies (ANAs), anti-cardiolipin antibodies, and rheumatoid factor (Jeremiah et al, 2014; Liu et al, 2014; Munoz et al, 2015; Chia et al, 2016; Fremond et al, 2016; Picard et al, 2016; Konig et al, 2017). Constitutive activation of STING, with the downstream activation of tank-binding kinase-1 and nuclear factor-κB, leads to raised levels of type I interferons and various cytokines and chemokines (Ishikawa et al, 2009; Abe et al, 2013; Gao et al, 2013; Xiao & Fitzgerald, 2013). SAVI is relatively refractory to glucocorticoids; however, partial responses to Janus kinase (JAK) inhibitors have been observed (Munoz et al, 2015; Fremond et al, 2016; Konig et al, 2017).

Murine models for SAVI and other autoinflammatory syndromes will facilitate studies of disease pathogenesis and the development of therapeutic strategies. Herein, we have generated a model for SAVI via the transgenic expression of a SAVI-associated *hSTING* mutation (N154S) in murine hematopoietic cells. Similar to SAVI (Liu et al, 2014; Fremond et al, 2016; Konig et al, 2017), *hSTING-N154S* transgenic mice exhibited the following: acral necrosis, dermal

---

[1]Department of Biochemistry and Molecular Biology, Cumming School of Medicine, University of Calgary, Calgary, Canada [2]The McCaig Institute for Bone and Joint Health, Cumming School of Medicine, University of Calgary, Calgary, Canada [3]Department of Pathology and Laboratory Medicine, Cumming School of Medicine, University of Calgary, Calgary, Canada [4]Department of Critical Care Medicine, Cumming School of Medicine, University of Calgary, Calgary, Canada [5]Calvin, Phoebe and Joan Snyder Institute for Chronic Diseases, Cumming School of Medicine, University of Calgary, Calgary, Canada [6]Auckland Cancer Society Research Centre, Faculty of Medical and Health Sciences, The University of Auckland, Auckland, New Zealand

Correspondence: marting@ucalgary.ca; jirik@ucalgary.ca

infiltrates, myositis, vasculopathy, lymphopenia, and elevated proinflammatory mediators and type I interferons. Unlike humans with activating mutations of STING (including the N154S *hSTING* mutation), *hSTING-N154S* mice failed to develop significant lung pathology. Importantly, and in keeping with constitutive STING activation being classified as an interferonopathy, the observed phenotype failed to develop in *hSTING-N154S* mice lacking the type I interferon α receptor subunit 1 (*ifnar1*).

# Results

## Gross morphological abnormalities of *hSTING-N154S* mice

By 8–10 wk of age, three of the five *hSTING-N154S* founder lines exhibited growth impairment, a failure to gain weight, and a reduced lifespan as a result of complications associated with the disease (Fig 1A–C). However, overall survival could not absolutely be determined in our *hSTING-154S* mice as the time to endpoint (e.g., sacrifice because of the severity of disease) was somewhat variable. We also observed that the disease in these three lines affected males and females equally. To reduce variability, we selected the 1,501 line (the most severe phenotype), and herein, all experimental observations are centered on this line only. In addition, all three lines developed progressive paw swelling (Fig 1D–F), accompanied by acral necrosis that was manifested by losses of ear cartilage as well as tail inflammation and shortening (Fig 1G and H). The progressive paw swelling that occurred in the three lines demonstrates that this was not the result of a gene insertion site defect (Fig S1A).

## Paw inflammation in *hSTING-N154S* mice

In contrast to WT paws (Fig 2A), *hSTING-N154S* paws exhibited edema and dense inflammatory cell infiltration of the dermis (Fig 2B), with areas of necrosis, including bone marrow necrosis (Fig 2B and C). A prominent inflammatory myositis, accompanied by muscle fiber loss, was invariably present (Fig 2C–E). There were only rare foci of pulmonary infiltrates (Fig S1B and C) and mild hind foot joint synovitis along with synovial lining cell hyperplasia and hypertrophy (Fig S1D), whereas proximal muscles only showed rare foci of infiltrates in interstitial areas (Fig S1E). We did not find evidence of inflammatory infiltrates or tissue necrosis in our surveys of other mouse tissues.

## Paw vasculopathy in *hSTING-N154S* mice

X-ray microscopy (XRM) imaging of Microfil$^R$-perfused *hSTING-N154S* mouse forepaws revealed dilation of large draining veins, often containing defects consistent with sizable venous thrombi, as well as multiple sites of small arterial and venous vessel stenoses and occlusions (Fig 3A–C). Consistent with the XRM imaging, there was histopathological evidence of paw vessel inflammation and damage (Fig 3D), as well as arteriolar lumenal occlusions by organizing bland thrombi (Fig 3E and F). We did not find convincing evidence of internal elastic lamina disruption that would be typical of a transmural vasculitis; hence, the findings were compatible with

the diagnosis of a vasculopathy. We did not find evidence of vessel occlusions or tissue necrosis in our surveys of other mouse tissues.

## Human *STING* expression in whole splenic tissue and selected cell populations

To examine mutant *hSTING* expression in the various splenic populations, including CD3$^+$ (T cells), CD11b$^+$ (macrophages), and CD19$^+$ (B cells), as well as CD31$^+$ endothelial cells that were isolated from the lung, we used a human-specific STING fluor-conjugated antibody (Fig S2A and B). When STING expression was assessed in the various cell populations derived from the spleen, for example, CD3$^+$ (T cells), CD11b$^+$ (macrophages), and CD19$^+$ (B cells), we discerned that only the transgene-positive cells expressed the human STING (Fig 4A–C). Percentages were relatively low possibly owing to (i) technical reasons associated with the efficiency of the intracellular staining process in different cells types; (ii) expression levels per cell being below the detection threshold of this method; and (iii) the possibility of variegated transgene expression. As there was a possibility that the Vav1 promoter could have resulted in the expression of mutant *hSTING* in the endothelium, we isolated CD31$^+$ CD41$^-$endothelial cells from the lung; human STING protein was not detected in isolated endothelial cells (Figs 4C and S2A). We also examined the splenic protein expression of STING in WT, *hSTING-154S*, and *hSTING-N154S* mice that had been crossed onto an *mSting*-KO background. We observed no significant increases in splenic STING expression in any of the mice that expressed the *hSTING-N154S* transgene as compared with WT mice (Fig 4D). This was due to the relatively low levels of transgene-derived mutant STING expression, best illustrated when *hSTING-N154S* mice were crossed onto an *mSting*-KO background. As expected, *mSting* protein expression was absent in the spleens of *mSting*-KO mice (Fig 4D).

## Lymphopenia in lymphoid tissues of *hSTING-N154S* mice

Because lymphopenia is a feature of SAVI, we investigated whether this would be reflected in the lymphoid tissues of *hSTING-N154S* mice. CD4$^+$ and CD8$^+$ abundance and ratios were, thus, determined for spleen, thymus, and lymph nodes of *hSTING-N154S* mice, their WT littermates, and *mSting*-KO mice. We found a marked reduction in the number of CD4$^+$ and CD8$^+$ cells in the spleen and lymph nodes of *hSTING-N154S* mice, but no differences were observed in the thymus. No significant differences in the percentages of these populations were seen when WT and *mSting*-KO mice were compared (Fig 5A–C).

## Serum type I interferons in *hSTING-N154S* mice

We also examined whether the *hSTING-N154S* phenotype was accompanied by the production of type I interferons. In RNA derived from the spleen, we found that *IFNβ* transcripts were modestly increased in *hSTING-N154S* mice as compared with WT littermates (Fig 5D). In addition, using a 13-plex Luminex assay, we found that the IFN-β levels were elevated in the sera of *hSTING-N154S* mice (Fig 5E), and using an ELISA, we were able to detect multiple murine

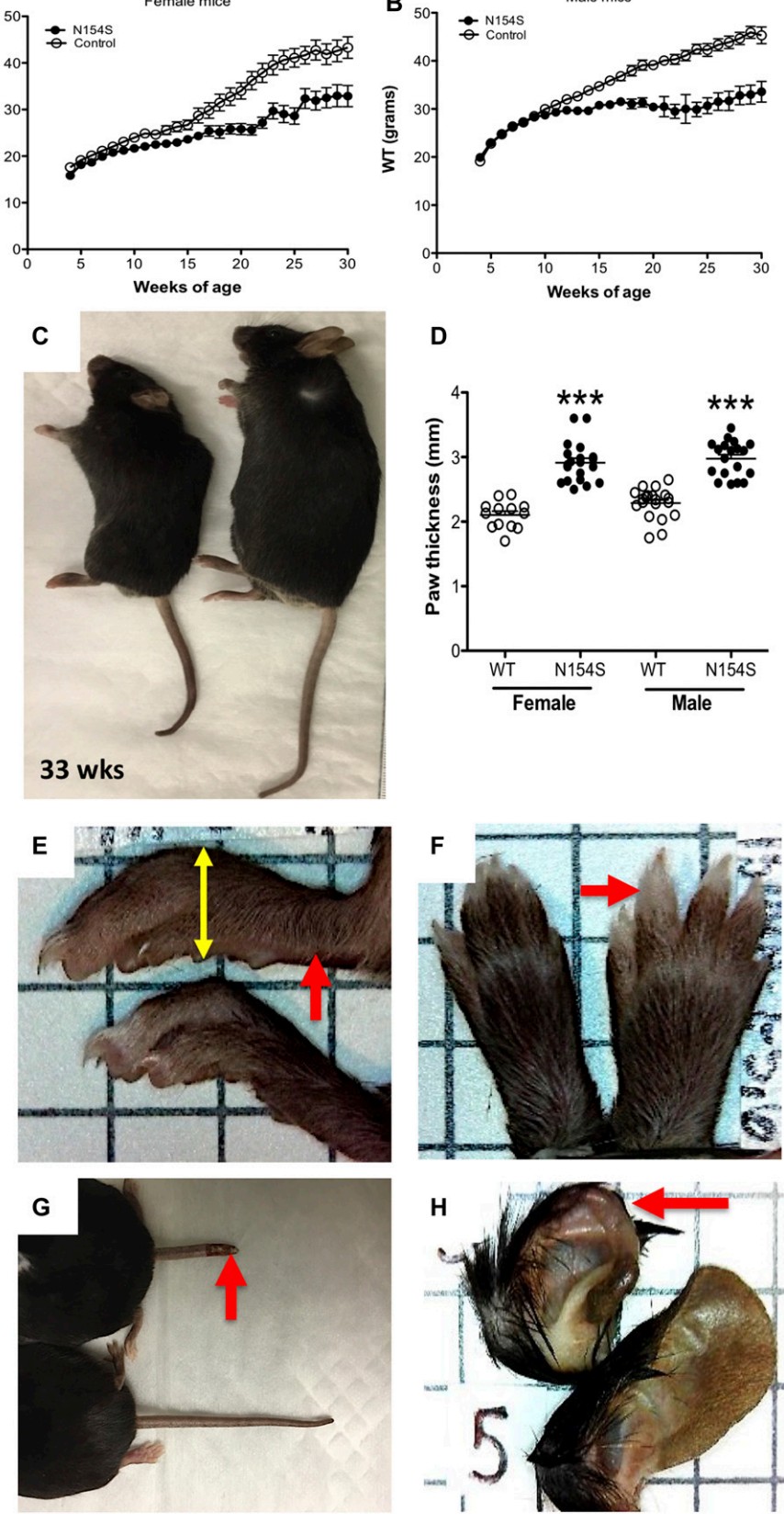

**Figure 1. *hSTING-N154S* mice show impaired weight gain, paw swelling, and acral necrosis.**
**(A, B)** Both female and male *hSTING-N154S* mice demonstrated a failure to gain weight starting at 8–10 wk of age. As discussed in the Results section, the "n" of mice used to calculate each time point was variable as all mice did not survive to endpoint (e.g., sacrifice due to the severity of disease); WT littermates were euthanized at these same time points as controls. **(C)** Generalized growth impairment was seen in *hSTING-N154S* mice (left) relative to WT littermates (right). **(D–F)** *hSTING-N154S* mice also developed progressive paw swelling that was first evident by ~6 wk of age (red arrows). Paw thickness was determined by dorsoventral measurement (yellow arrow) using digital calipers. **(G, H)** *hSTING-N154S* mice developed tail inflammation and swelling with ensuing necrosis that lead to tail shortening (E). **(H)** These mice also exhibited losses of ear cartilage. For the paw thickness data, a one-way ANOVA with Tukey's multiple comparisons post hoc test was used. ***$P < 0.001$ versus WT, n ≥ 13 per group.

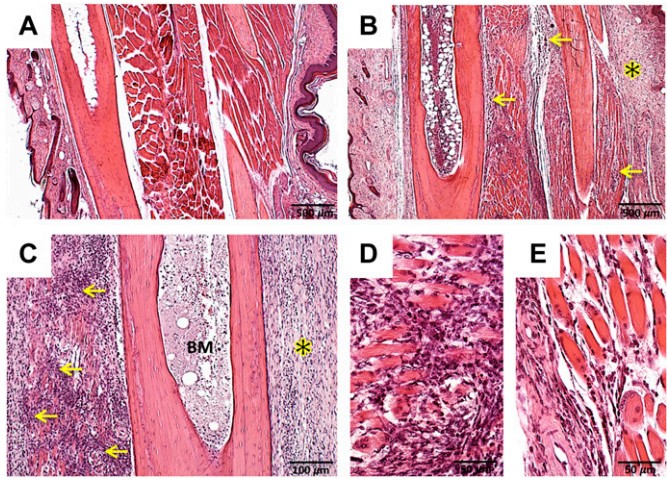

**Figure 2. Paw inflammation in *hSTING-N154S* mice.**
**(A, B)** Sections of representative hind paw digits from a WT littermate (A) and an *hSTING-N154S* mouse (B). The latter shows evidence of dermal edema, inflammatory infiltrates, and a region of necrosis (yellow asterisk). Infiltrates are also evident in skeletal muscle (arrows); there was also increased inflammatory cell accumulation within the bone marrow. **(C)** Higher magnification view of an *hSTING-N154S* digit showing marked myositis (arrows), dermal inflammatory infiltrate with edema (asterisk), and bone marrow necrosis (BM). **(D, E)** Paw inflammation in two different *hSTING-N154S* mice that had marked myositis associated with prominent inflammatory cell infiltrates, edema, and muscle fiber loss. Hematoxylin and eosin staining were used. Magnification: (A, B) 100×; (C) 200×; (D, E) 400×.

IFN-α variants (1, 2, 4, 5, 6) and significant increases of IFN-α in *hSTING-N154S* sera (Fig 5F).

### Serum hyper-cytokinemia in *hSTING-N154S* mice

Compared with littermate control sera, serum samples from *hSTING-N154S* mice contained elevated levels of several chemokines (CCL2, CCL3, CCL4, CCL5, CXCL1, CXCL9, and CXCL10) and cytokines (TNF-α, IL-6, G-CSF, and IL-5) (Fig 6). Aside from IL-6 and IL-5, no significant differences were observed among the other interleukins that were tested (Fig S3).

### Pulmonary and peripheral blood lymphopenia in *hSTING-N154S* mice

Lymphopenia was present in *hSTING-N154S* peripheral blood (Fig 7A), with decreases in CD19[+], CD4[+], and CD8[+] cells as compared with littermate controls (Fig 7C). In contrast, Ly6G[+] cells (consisting of neutrophils) were significantly elevated in the *hSTING-N154S* animals. The same effect on lymphocytes was also evident in the analysis of dissociated lung tissue (Fig 7B), although the decrease in CD19[+] cells was not statistically significant (Fig 7D). Alveolar macrophage (F4/80[+] CD11b[int]) levels, quantified in both lung tissue and bronchioalveolar lavage (BAL) fluid, varied considerably between mice, with no significant differences being observed between littermate controls and *hSTING-N154S* mice (Fig S4). Furthermore, lymphocytes and Ly6G[+] cells were not detected in the BAL fluid of mice from either group (data not shown).

### ANAs in *hSTING-N154S* mice

Since ANAs have been observed in human SAVI, we undertook an analysis of *hSTING-N154S* and littermate control sera (Figs S5 and S6). This revealed that 15 of 19 transgenic animals were ANA[+], with titers varying between 1:160 and 1:1,280. A proportion (4 of 11) of littermate controls also were ANA[+], albeit at titers of 1:320 or less (3 of 4), with only one animal having a titer of 1:1,280. Interestingly, one of the transgenic sera also contained reactivity towards Jo-1, PL-7, and SRP, markers associated with human autoimmune myositis and/or interstitial lung disease (Benveniste et al, 2016).

### Phenotype *hSTING-N154S* mice depends on IFNAR1

To determine whether the observed phenotype required intact type I IFN receptor signaling, we interbred *ifnar1*-KO (C57BL/6) and N154S (C57BL/6) mice to place the *hSTING-N154S* transgene onto an *ifnar1*-KO background. As before (Fig 1A–C), the *hSTING-N154S* offspring were smaller than either the age-matched (13-22 wk-old) littermate controls or the *hSTING-N154S/ifnar1*-KO mice (Fig 8A). Importantly, the *hSTING-N154S/ifnar1*-KO mice were indistinguishable from the WT littermate controls and failed to develop evidence of acral necrosis or the marked paw swelling characteristic of *hSTING-N154S* mice (Fig 8A–C). In keeping with this result, histological examination of *hSTING-N154S/ifnar1*-KO paws revealed no evidence of dermal inflammation, necrosis, or myositis (Fig S7A). Body weights of *hSTING-N154S/ifnar1*-KO mice (37.7 ± 3.83 g) were similar to those of WT mice (36.3 ± 2.00 g) and differed significantly from those of age-matched *hSTING-N154S* mice (25.5 ± 0.56 g, *P* < 0.05) (Fig 8D). Caliper measurements of hind paws also showed no differences between the *hSTING-N154S/ifnar1*-KO and WT littermates. In contrast, the hind paws of the *hSTING-N154S* mice were ~30% thicker than those of both WT (*P* < 0.05) and *hSTING-N154S/ifnar1*-KO mice (*P* < 0.01) (Fig 8E). Splenic enlargement observed in *hSTING-N154S* mice was not observed in the *hSTING-N154S/ifnar1*-KO mice (1.2 ± 0.05 mg/g versus *hSTING-N154S* 3.6 ± 0.19 mg/g, *P* < 0.01) (Fig 8F). Last, we found that the *hSTING-N154S/ifnar1*-KO mice did not have elevated serum levels of cytokines and chemokines that were found in N154S mice (Fig 9).

## Discussion

We have generated a mouse model of human SAVI by expressing a constitutively active human STING mutant in hematopoietic cells. Despite mutant STING expression being restricted to the hematopoietic compartment, our *hSTING-N154S* mice exhibited many of the characteristics that have been observed in SAVI. These similarities include growth failure, dermal inflammation, acral necrosis with tissue loss due to a vasculopathy with vessel thrombosis, myositis, increased proinflammatory cytokine/chemokine accumulation, and lymphopenia (Jeremiah et al, 2014; Liu et al, 2014; Munoz et al, 2015; Omoyinmi et al, 2015; Chia et al, 2016; Fremond et al, 2016; Picard et al, 2016; Konig et al, 2017). For example, Liu et al (2014) reported that serum concentrations of several proinflammatory cytokines, such as CXCL10 and TNFα, were significantly

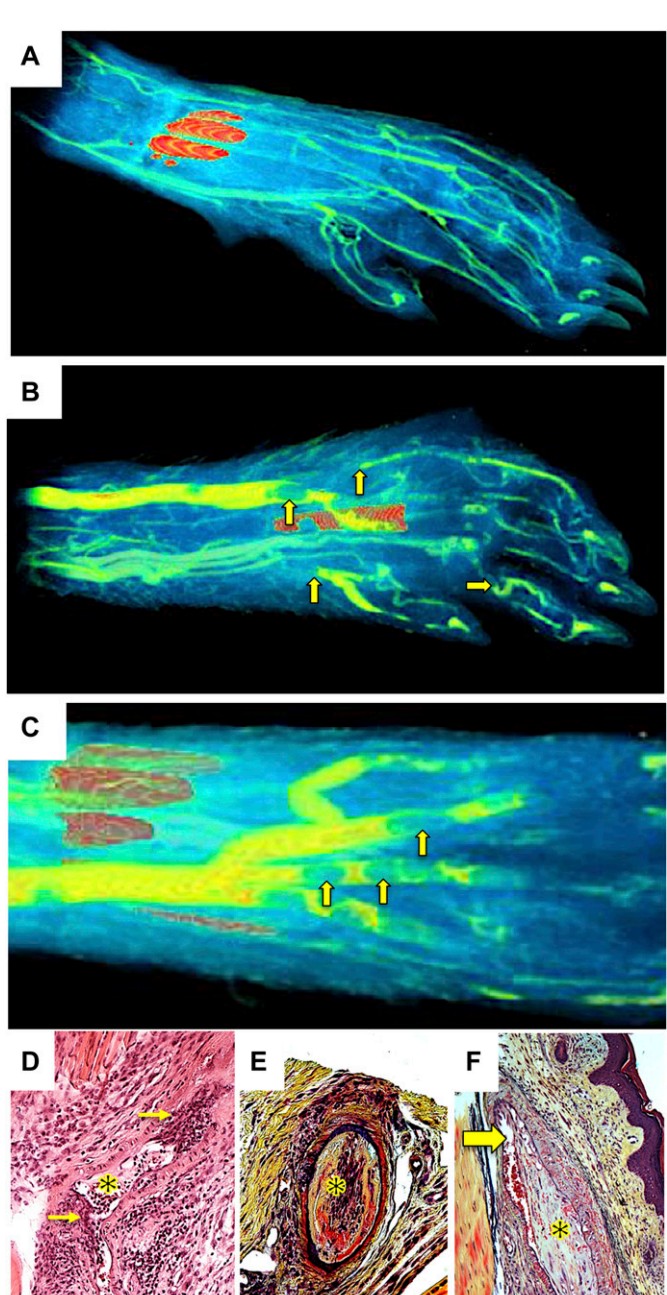

**Figure 3. Multiple arterial and venous thromboses in the paws of *hSTING-N154S* mice revealed by XRM and histopathology.**
**(A–C)** Representative XRM imaging of Microfil[R]-perfused forepaws from (A) a WT littermate and (B) an *hSTING-N154S* transgenic mouse showing venous dilation, thrombi, and multiple sites of vessel interruption (arrows). **(C)** Higher magnification view of the dilated veins in a forepaw from a transgenic mouse shows multiple venous thrombi (arrows). **(D)** Disrupted arteriole with transmural inflammatory infiltrates (arrows) and luminal fibrin deposition (asterisk). Note the absence of red blood cells in the lumen. **(E)** Paw arteriole showing complete occlusion of the lumen with collagenous (yellow) organization (asterisk) of the thrombus and residual fibrin (red). **(F)** Organizing paw arteriolar thrombosis (asterisk) showing a residual cleft of lumen containing red blood cells (arrow). This thrombus is older than the one in (D) with more mature collagen (green-yellow). **(E, F)** Internal elastic laminae were intact with no evidence of transmural

elevated, as they were in the *hSTING-N154S* mice we generated. Interestingly, and in contrast to myositis models requiring immunization with myosin protein plus adjuvant (Allenbach et al, 2009; Kang et al, 2015), *hSTING*-N154S mice spontaneously developed severe myositis of interossei muscle.

XRM imaging of *hSTING-N154S* paws, together with histological analysis, revealed evidence of a severe vasculopathy. Thus, after perfusion of the mice with a radio-opaque monomer that polymerizes in the cold, XRM was used to obtain high-resolution images of the vasculature. This revealed that *hSTING-N154S* paws contained widespread stenoses and obstructions of both arterioles and venules, together with the presence of prominent thrombi in large veins. Histological demonstration of arteriolar lumen occlusions by bland organizing clots was also obtained. An as of yet unresolved question revolves around the reason(s) for the acral distribution of pathology in the *hSTING-N154S* mice. The predicted reduced temperature of extremities (tail, ears, and paws) might be an etiological factor. In this regard, it is interesting that the acral lesions in SAVI are aggravated by cold weather, raising the possibility that cryoprotein(s) might be an etiological factor in the pathology (Munoz et al, 2015; Picard et al, 2016; Konig et al, 2017). This possibility, or perhaps the small vessel vasculopathy in combination with cold-induced vasoconstriction, might account for the chilblains, Raynaud phenomenon, and livedoid rashes seen in SAVI (Stoffels & Kastner, 2016), although we were not able to visualize any cryoprecipitates after prolonged cooling of *hSTING*-N154S sera (data not shown). Reducing the temperature of the mouse holding room is a possibility because this could aggravate or accelerate disease progression in the *hSTING-N154S* mice. However, further studies are required to determine whether cryoproteins, and/or factors associated with anti-phospholipid syndrome, are present in these mice.

Because various STING mutations have been reported to result in interstitial pulmonary inflammation and fibrosis (Jeremiah et al, 2014; Liu et al, 2014; Picard et al, 2016), we examined the lungs of *hSTING-N154S* mice. Unlike humans with activating *STING* mutations, *hSTING-N154S* mice failed to develop significant lung inflammation or fibrosis. Only very rare foci of hematopoietic cell infiltrates were present, even in mice >6 mo of age. This was consistent with our finding that the lungs and BAL fluid of *hSTING-N154S* mice did not show significant increases in hematopoietic cell numbers, except for neutrophils in the former. Because *hSTING-N154S* expression is confined to hematopoietic cells, the absence of significant pulmonary disease in the mice suggested that expression of constitutively active STING protein in lung parenchymal cells may be required for development of interstitial lung disease and fibrosis.

Regarding lung involvement, Warner et al (2017) described the phenotype of mice having an N153S knock-in mutation of *mSting*. These mice developed severe lung inflammation, nonacral skin ulceration, as well as hyper-cytokinemia and lymphopenia (Warner et al, 2017). Similar to the *hSTING-N154S* mice, 4–6-mo-old *mSting-N153S* mice had elevated serum proinflammatory mediators (Warner et al, 2017), albeit at lower levels than that of *hSTING-N154S* mice. The

vasculitis. **(A–C)** The orange areas in the paws are the result of incomplete decalcification. Stains: hematoxylin and eosin (D) and Movat pentachrome (E, F). Magnification: 400×.

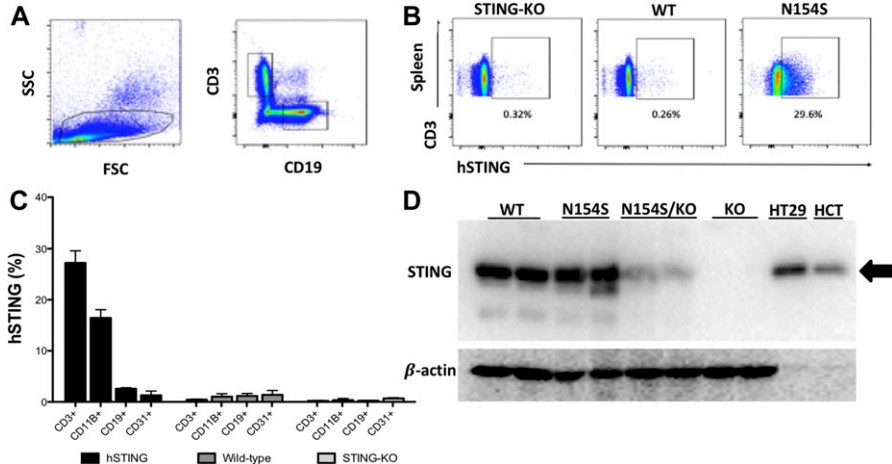

**Figure 4. Western blot and intracellular staining show mutant human STING expression in splenic tissue and splenic cell populations.**
**(A)** Representative dot plots showing the initial gating settings for the population of cells that were selected for FACS analyses (left panel) and the relative proportion of CD3+ and CD19+ cells (right panel) in a single-cell suspension of dissociated spleen. **(B)** Representative dot plots to show the percentage of CD3+ hSTING+ cells from the spleen of *mSting*-KO, WT, and *Vav1-hSTING-N154S* mice. Numbers below each gate are the percentage of cells within the corresponding gate. **(C)** Representative histogram showing human STING expression in the various splenic populations. CD3+ (T cells), CD11b+ (macrophages), and CD19+ (B cells) were obtained from the spleen; CD31+ endothelial cells were isolated from the lung. **(D)** Western blot detection of m/hSTING expression in splenic lysates using a polyclonal antibody that recognizes both mouse and human STING as described in the Results section. As positive controls, two human CRC lines known to express STING protein were used: HT29 and HCT116 (HCT). For spleen analyses, 40 µg of protein/lane and for CRC cell protein, 10 µg/lane were loaded. Arrow indicates the STING protein band in the human CRC lines.

*mSting-N153S* mice were not reported to develop acral inflammation and necrosis, vasculopathy, or myositis. Also, deletion of *Irf3* did not block the phenotype of these mice, likely owing to redundancy in the pathways involved in mediating the type I interferon responses. Modest increases in interferon-stimulated genes were seen when fibroblasts from the *mSting-N153S* mice or from humans with *STING-N154S*–associated SAVI were evaluated (Warner et al, 2017). Similarly, we found increases in both IFN-$\alpha$ and IFN-$\beta$ in *hSTING-N154S* sera, as well as increased levels of CXCL10, a marker that often accompanies interferon production (Luster & Ravetch, 1987; Vanguri & Farber, 1990).

Bouis et al (2018) recently reported the generation of mice having a V154M knock-in of *mSting*. These mice demonstrated an increased mortality rate, weight loss, and evidence of both lung and renal hematopoietic cell infiltrates. They also developed pronounced lymphopenia, resulting in a SCID-like phenotype with hypo-$\gamma$-globulinemia and NK cell depletion. The reported phenotype did not describe acral inflammation and necrosis, vasculopathy, or myositis. Interestingly, the SCID-like phenotype was not reversed by interbreeding *mSting-V154M* mice with *Ifnar1*-knockout mice, although the inhibitory effect of mutant *mSting* activation on T cells was partially reversed. These results are consistent with other reports (Cerboni et al, 2017), and our unpublished in vitro observations, indicating that the negative effects of STING activation on T cells is relatively independent of an autocrine type I interferon effect. Furthermore, agonist-mediated *mSting* protein activation was shown to be toxic to mouse B lymphocytes (Tang et al, 2016). Lastly, Motwani et al (2019) developed two *mSting*-mutant knock-ins that developed similar phenotypic features as the previous two *mSting* mutant knock-ins, although these too did not develop acral necrosis (Motwani et al, 2019). They also found that the phenotype was present in the absence of the type I IFN receptor. Similar to these various *mSting* knock-ins, we also observed significant decreases in lymphocytes in the peripheral blood, lungs, spleen, and lymph nodes of *hSTING-N154S* mice. These findings are consistent with reports of peripheral blood lymphopenia in individual carrying SAVI mutations.

In view of these discrepancies, an important question remains: why is the phenotype of our transgenic model different from the three *mSting* knock-in transgenics that have not been reported to develop acral necrosis? One obvious possibility concerns the use of an ectopic gene promoter to drive *hSTING-N154S* expression. The Vav1 gene promoter is unlikely to be subject to the same regulation as the endogenous *mSting* gene promoter, which possibly could have led to higher-than-normal WT levels of mutant *hSTING* protein expression. However, upon analyses of total STING expression in splenic lysates, we found no significant increases in *Vav1-hSTING-N154S*–directed protein expression. Furthermore, when *hSTING-N154S* mice were placed on an *mSting*-KO background, relatively low levels of STING expression were seen using an antibody that detects both human and murine STING. Although it is possible that the constitutively active mutant *hSTING* protein undergoes rapid degradation, another possibility, as suggested by the disease-attenuating effect of antibiotic treatment reported for the *mSting-V154M* mouse (Bouis et al, 2018), is that differences in microbiota between the various transgenics might account, at least in part, for their phenotypic variability.

Why do our transgenics invariably develop prominent paw inflammation and acral necrosis? It has been reported that the Vav1 promoter may be expressed in endothelial cells (Joseph et al, 2013) and thus may have been responsible for the observed vascular pathology. Although it is possible that mutant hSTING protein expression levels were below the ability of the intracellular detection method we used, hSTING expression in the endothelial cells that had been isolated from the lung was undetectable. Furthermore, if acral necrosis was indeed dependent on ectopic Vav1 promoter-directed mutant *hSTING* expression in the endothelium, why would the *mSting*-mutant knock-ins lack endothelial expression, given that *mSting* is thought to be ubiquitously expressed?

Unlike the *mSting* mutant knock-in models (Warner et al, 2017; Bouis et al, 2018; Motwani et al, 2019), we did not observe any significant lung inflammation. One possibility is that mutant STING expression in the lung parenchymal cells is required. Because

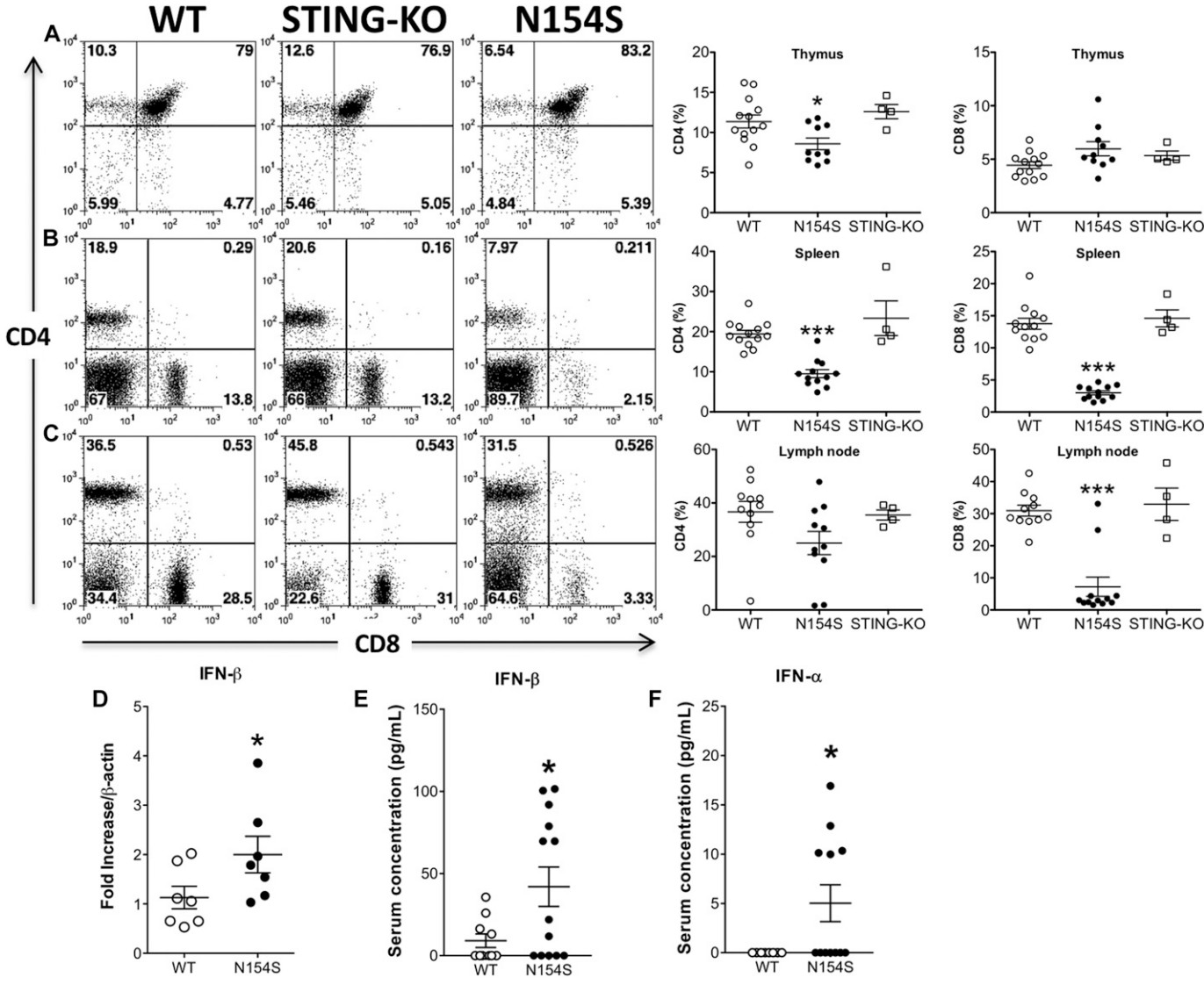

**Figure 5. T-cell lymphopenia and type I interferon levels in *hSTING-N154S* mice.**
**(A–C)** Whereas CD4[+] T cell numbers were moderately reduced in the thymi of *hSTING-N154S* mice (A), there were marked reductions in the number of CD4[+] and CD8[+] cells in the spleens (B) and lymph nodes (C) when compared with WT mice. There were no differences in these T-cell populations when WT and *mSting*-KO mice were compared. One-way ANOVA with Tukey's multiple comparisons post hoc tests were used to analyze group differences. WT and *hSTING-N154S*, n ≥ 10; *mSting*-KO, n = 4. Horizontal lines represent the mean ± SEM with significant differences denoted as $*P < 0.05$ or $***P < 0.001$ versus WT. **(D)** Using quantitative RT-PCR analysis of splenic tissues, we observed that IFN-$\beta$ transcripts were modestly increased in the *hSTING-N154S* mice (n = 7) relative to those in WT littermates (n = 7). **(E)** 13-plex Luminex assay of serum showed that mIFN-$\beta$ levels were elevated in the sera of 8 of 13 *hSTING-N154S* mice (n = 13) compared with 4 of 10 WT littermates (n = 10). **(F)** Compared with WT littermates, there was also a significant increase in mIFN-$\alpha$ levels as detected via ELISA in the sera of *hSTING-N154S* mice (n = 12) (LumiKine Xpress mIFN-$\alpha$ ELISA kit). Horizontal lines represent the mean ± SEM serum concentrations (pg/ml) of murine IFN-$\beta$ or IFN-$\alpha$. Data are pooled from five independent experiments (n = 1–5 for each group). Unpaired $t$ test was carried out between *hSTING-N154S* and WT groups where $*P < 0.05$.

*Vav1-hSTING-N154S* expression is primarily confined to hematopoietic cells, the lack of lung disease in our model suggests that the expression of constitutively active STING in lung parenchymal cells may be necessary for the lung inflammation to develop.

In addition to decreased T lymphocyte levels, there were reductions in peripheral blood and pulmonary B lymphocytes in *hSTING-N154S* mice, although in the lungs, this did not reach statistical significance. In contrast, there were increased pulmonary and peripheral blood Ly6G[+] cells (neutrophils) in *hSTING-N154S* mice compared with controls. Recently, Kim et al (2018) demonstrated

that B lymphocytes, via direct interaction with neutrophils in the lungs, facilitate the clearance of aging cells. Moreover, depletion of B lymphocytes resulted in the accumulation of aged PMNs within the lungs, which promoted fibrotic interstitial lung disease (Kim et al, 2018). B-cell lymphopenia may thus be contributing to the increased level of neutrophils seen in the *hSTING-N154S* samples. However, increased levels of mediators such as G-CSF, via their ability to increase bone marrow generation and mobilization of granulocytes (Bendall & Bradstock, 2014), may have also promoted neutrophil numbers.

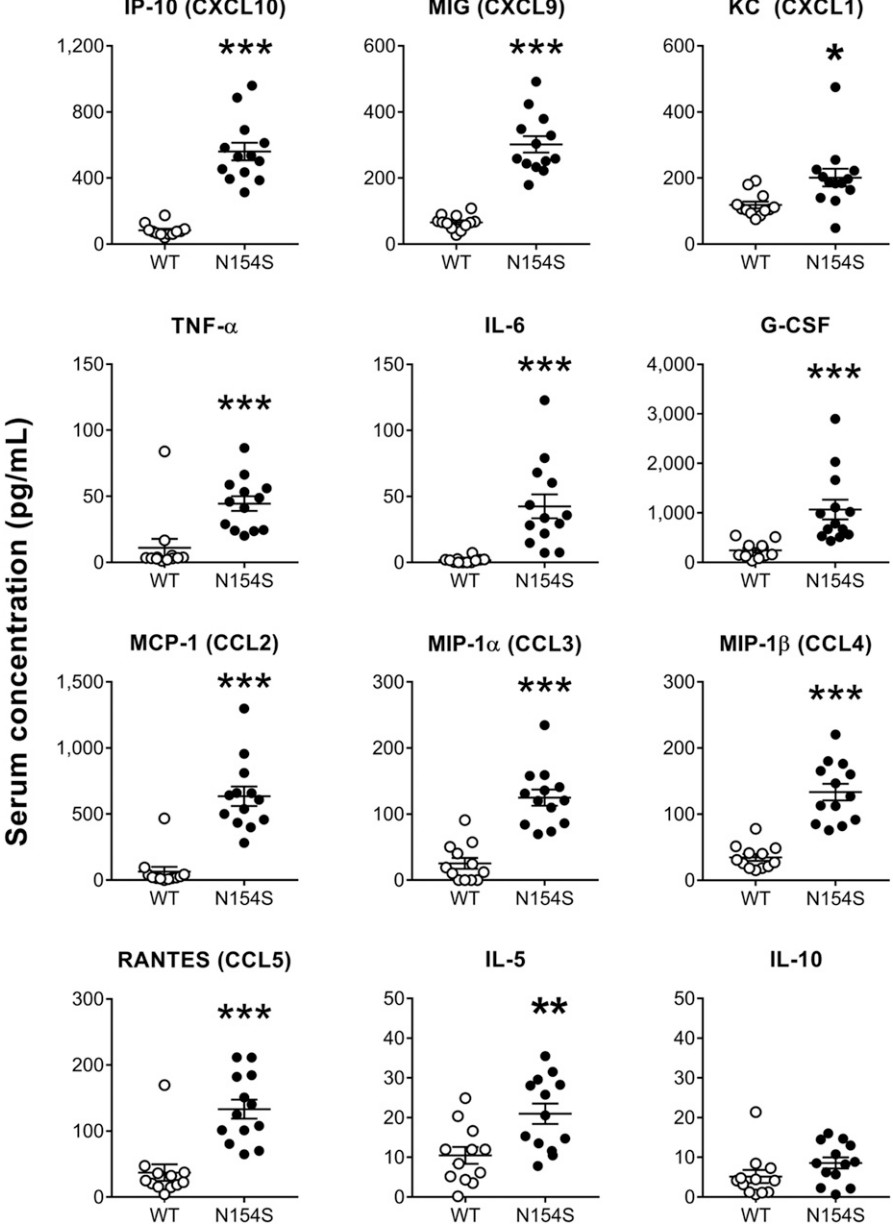

**Figure 6. Increased cytokine levels in *hSTING-N154S* mice.**
Serum cytokine levels in *hSTING-N154S* mice (n = 13) and WT littermate controls (n = 12) were measured by 31-plex murine cytokine array. Each symbol represents an individual mouse and horizontal lines represent the mean ± SEM of serum concentrations for each cytokine (pg/ml). Data are pooled from five independent experiments (n = 1–5 for each group). Unpaired *t* test was carried out between *hSTING-N154S* and WT groups where *P < 0.05, **P < 0.01, ***P < 0.001, and ****P < 0.0001. See Fig S3 for the remainder of the Luminex results from these mice.

Consistent with reports of positive ANA tests in SAVI, we found that 15 of 19 transgenic animals were ANA+ (with titers ranging between 1:160 and 1:1,280), whereas 4 of 11 of the transgene-negative littermate controls were ANA+ (titers of 1:320 or less). C57BL/6 wild-type mice are known to be ANA+ to varying degrees (Bygrave et al, 2004). Interestingly, one of the transgenic sera also contained reactivity towards Jo-1, PL-7, and SRP, markers associated in humans with autoimmune myositis and/or interstitial lung disease (Richards et al, 2009; Benveniste et al, 2016). A full breakdown of the autoimmune profiles of the *hSTING* mutant mice can be found in Fig S5.

Until recently, therapeutic treatment of vasculopathies has been relatively unsuccessful. However, the use of JAK inhibitors in vitro, such as tofacitinib, ruxolitinib, and baricitinib, has shown some promise as they reduced the transcription of *IFNB1* and several other interferon response genes in fibroblasts obtained from human patients (Liu et al, 2014). In the clinical setting, JAK inhibitors were shown to be of some therapeutic benefit owing to their ability to down-regulate the type I interferon receptor-initiated signal transduction pathway (Munoz et al, 2015; Fremond et al, 2016; Konig et al, 2017). IFNAR1, a member of the helical cytokine class II family of receptors, is a critical component of the IFN signaling pathway (Novick et al, 1994; Peng et al, 2015). This receptor activates intracellular signal transduction in response to all type I interferons, including, but not limited to, IFN-β and the various IFN-α subtypes (Garcia-Sastre & Biron, 2006). In keeping with the phenotype of the *hSTING-N154S* mice being dependent on type I interferons, we found that absence of *Ifnar1* prevented the development of the dramatic phenotype seen in the *hSTING-N154S* mice. Thus, paw

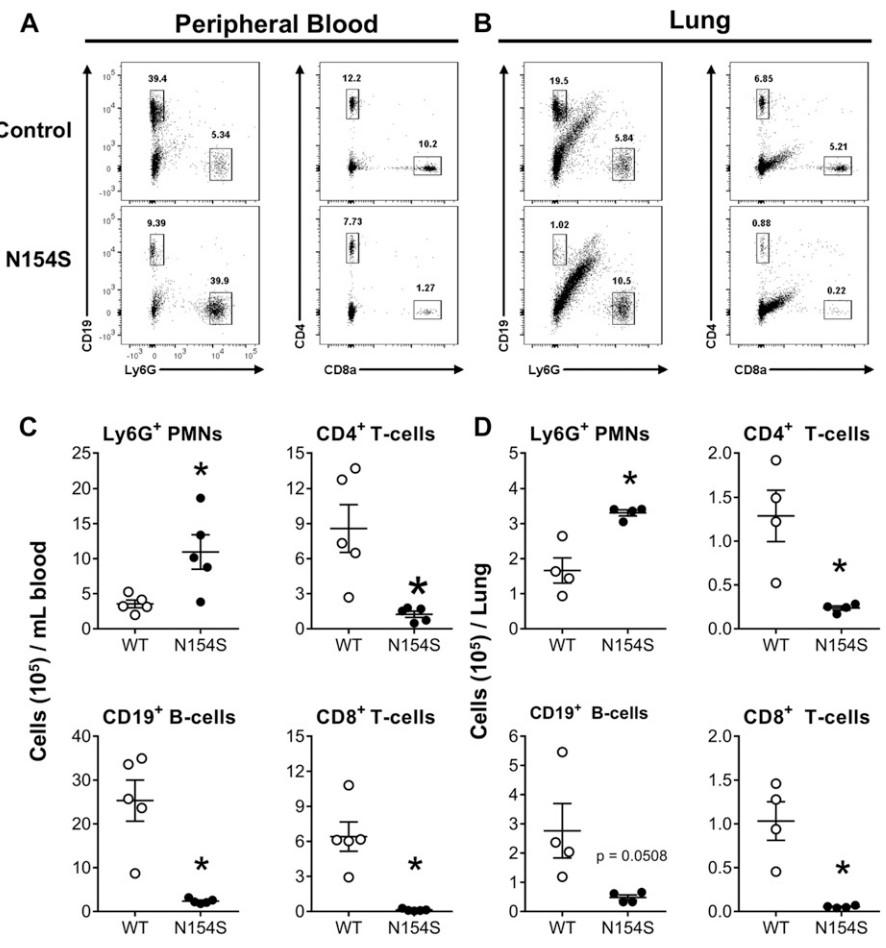

**Figure 7. Lymphopenia in *hSTING-N154S* mice.**
**(A, B)** Representative dot plots show gating and relative proportion of Ly6G+, CD19+, and CD4+ and CD8+ cells in single-cell suspensions of peripheral blood (A) and dissociated lung tissue (B) from *hSTING-N154S* mice and WT littermates that were analyzed by multicolour flow cytometry. Numbers above each gate are the percentage of cells within the corresponding gate. Regarding the intermediate CD19+/Ly6G staining (B), there are no known leukocytes that coexpress CD19 and Ly6G, and seeing that this population does not exist in the peripheral blood dot plots, this likely represents a lung parenchymal cell population. **(C, D)** Scatter plots showing the number of Ly6G+, CD19+, CD4+, and CD8a+ cells in peripheral blood (C) or dissociated lung tissue (D) from WT and *hSTING-N154S* mice. Each symbol represents an individual mouse and horizontal lines represent the mean ± SEM number of cells per mL of peripheral blood (C, n = 5) or per lung (D, n = 4) for each group. Data are pooled from two independent experiments (n = 1–3 for each group). Unpaired *t* test was used to compare between the two groups with *P*-values of <0.05 considered to be significant (\**P* < 0.05, \*\**P* < 0.01).

inflammation, acral necrosis, myositis, and proinflammatory cytokine/chemokine production was absent in *hSTING-N154S* mice lacking the *Ifnar1*.

Extrapolating to humans, our results suggest that IFNAR1 inhibition is likely to be of therapeutic benefit in SAVI. Indeed, the development and testing of anti-IFNAR1 antibodies such as anifrolumab (Peng et al, 2015; Furie et al, 2017) for use in humans is currently an active area of investigation and clinical trials. In this vein, work on the development of STING inhibitors also offers considerable promise (Haag et al, 2018). The SAVI model we have generated, based on the activity of a mutant human STING protein, represents not only a model system for dissecting mechanisms involved in the pathogenesis of SAVI but will also serve as a useful preclinical tool for the in vivo evaluation of therapeutics aimed at curtailing abnormal STING activity.

# Materials and Methods

### Mice

These studies were conducted in accordance with the guidelines of the Canadian Council of Animal Care, with all protocols approved by the Health Sciences Animal Care Committee of the University of Calgary. C57BL/6, Golden ticket (*TMEM173^{gt:gt}*) on a C57BL/6 background, and B6.129S2-*Ifnar1^{tm1Agt}*/Mm (backcrossed to C57BL/6 for at least five generations) were from Jackson Laboratories. WT refers to mice that were littermate controls for the transgenic animals being analyzed. Mice were fed standard laboratory chow, allowed water ad libitum, and maintained in independently ventilated micro-isolator units at 22 ± 1°C), 65–70% humidity, and a 12-h light/dark cycle.

### Generation of *hSTING-N154S* transgenic lines

Transgene generation involved inserting the *hSTING-N154S* mutant cDNA downstream of the murine *Vav1* gene promoter to obtain hematopoietic cell-specific expression. The 2.3/4.4(HS21/45) *Vav*-hCD4 (Clone#2) 11.2-Kb vector generously provided by Dr. Jerry Adams (The Walter and Eliza Hall Institute of Medical Research, Melbourne, Australia) (Ogilvy et al, 1999) was used (Fig S7B). The *Vav1*-hCD4 construct was digested with *Sfi*I and *Not*I to remove the hCD4 cDNA, and this was replaced with a synthetic (Celtek) cDNA encoding the consensus *hSTING* sequence plus the N154S point mutation reported in Liu et al (2014). The *Sfi*I site within the mutant *STING* cDNA was eliminated via codon substitution to facilitate cloning of the cDNA

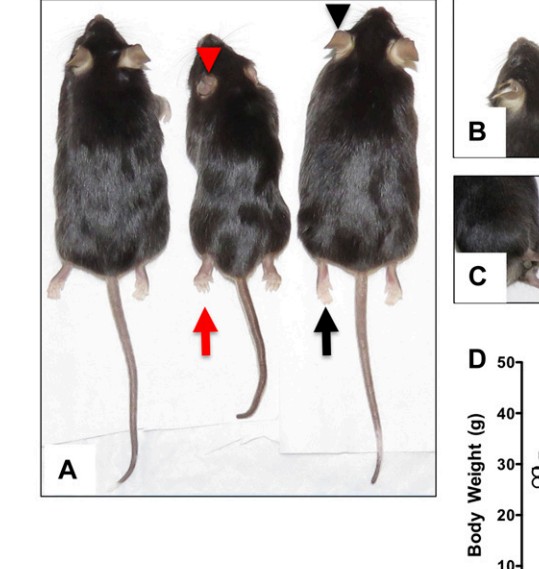
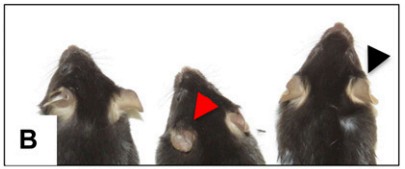
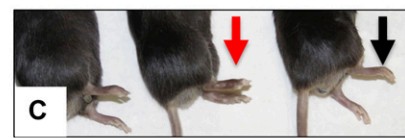
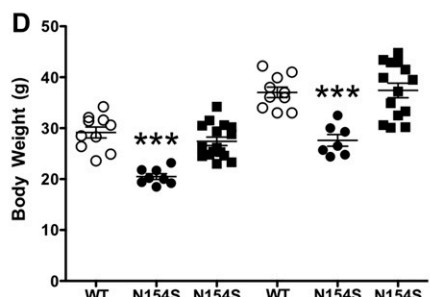
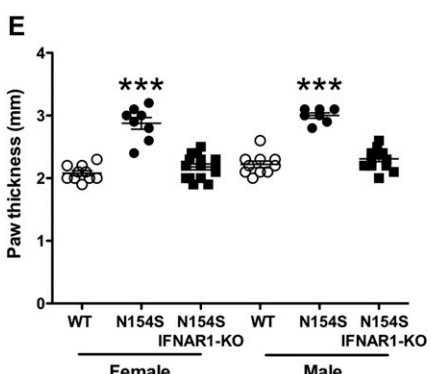
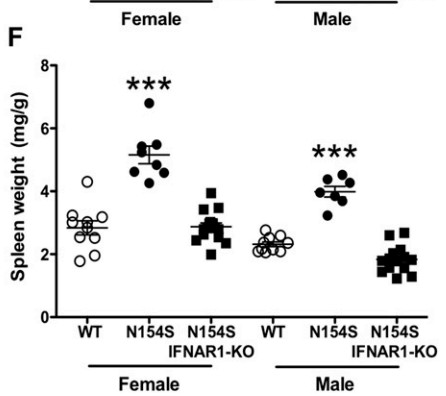

**Figure 8. *hSTING-N154S* vasculopathy is prevented by the lack of *ifnar1*.**
**(A–C)** Representative photographs of male age-matched WT (left), *hSTING-N154S* (center), and *hSTING-N154S/ifnar1*-KO (right) mice showing differences in body size (A), ear pathology (B), and paw thickness (C). Arrows and arrowheads indicate differences in the hind paws and ears, respectively, of *hSTING-N154S* mice either in the presence (red) or absence (black) of *ifnar1*. **(D–F)** Scatter plots showing the differences in body weight (D), paw thickness (E), and spleen weight (F) amongst WT (open circles), *hSTING-N154S* (closed circles), and *hSTING-N154S/ifnar1*-KO (closed squares) male and female mice. **(D–F)** Each symbol represents an individual mouse (D–F), and horizontal lines represent the mean ± SEM. Spleen weight (F) was normalized to body weight (mg/g of body weight). **(D–F)** Statistical significance between data sets (D–F) was assessed by one-way ANOVA followed by Tukey's multiple comparisons post hoc test between all groups. Significant differences between WT (M/F, n = 10) and *hSTING*-N154S (M, n = 7; F, n = 8) or between *hSTING-N154S* and *hSTING-N154S/Ifnar1*-KO (M, n = 14; F, n = 16) mice are denoted by ***$P < 0.001$.

into the *Vav1* backbone using a *Sfi*I/*Not*I digestion (Fig S7C). Bacteria carrying the plasmid were grown in Luria broth (244620; BD Difco) with 100 µg/ml ampicillin (A9518; Sigma-Aldrich) and purified with the PureLink HiPure Plasmid Midiprep Kit (K210015; Invitrogen). The DNA fragment containing the transgene (Fig S7B) was removed from the vector by *Hind*III digestion and purified using Promega's Wizard SV Gel and PCR Clean-Up System A9281. Transgenic lines were produced via pronuclear microinjection of the *Vav1* gene promoter-*hSTING-N154S* construct into C57BL/6 × DBA F1 embryos at the University of Calgary's Clara Christie Centre for Mouse Genomics. Founders were identified using the following primers: SOEcolchF 5'-GGC GGT GGT GAA GGA ACG AG-3' and SOEcolchR 5'-CCT TGA TGC CAG CAC GGT CA-3', 5% DMSO with a cycling program of 95°C 3 min (95°C 15 s, 69°C 15 s, and 72°C 60 s) × 35 s, 72°C 5 min, using the KAPA (D-MARK KK7352) Hot Start genotyping system. Five *hSTING*-N154S founders were identified and three of these that showed paw swelling had been backcrossed onto the C57BL/6 background for between five and eight generations during these experiments. All transgenic mice used were hemizygous for the transgene.

## Histology

Tissue samples were fixed in 10% neutral buffered formalin, embedded in paraffin, with 4-µm sections before staining. See figure legends for specific stains used.

## Real-time PCR

Spleens from *hSTING-N154S*, WT, and *mSting*-KO mice (Golden-ticket, *gt:gt*) were mechanically disrupted and lysed in QIAzol lysis reagent (QIAGEN) before chloroform and isopropanol RNA extraction. For cDNA synthesis, 1 µg of RNA was treated with DNAse (Promega) followed by RT-PCR with 10-mM dNTPs, random primers, and Superscript II reverse transcriptase (Invitrogen). Real-time PCR of cDNAs was carried out using the LightCycler FastStart DNA MasterPLUS SYBR Green Kit (Roche). Data were normalized to β-actin mRNA and experimental transcripts expressed as the relative fold-change in mRNA compared with controls. Primer sequences were as previously used (Downey et al, 2014).

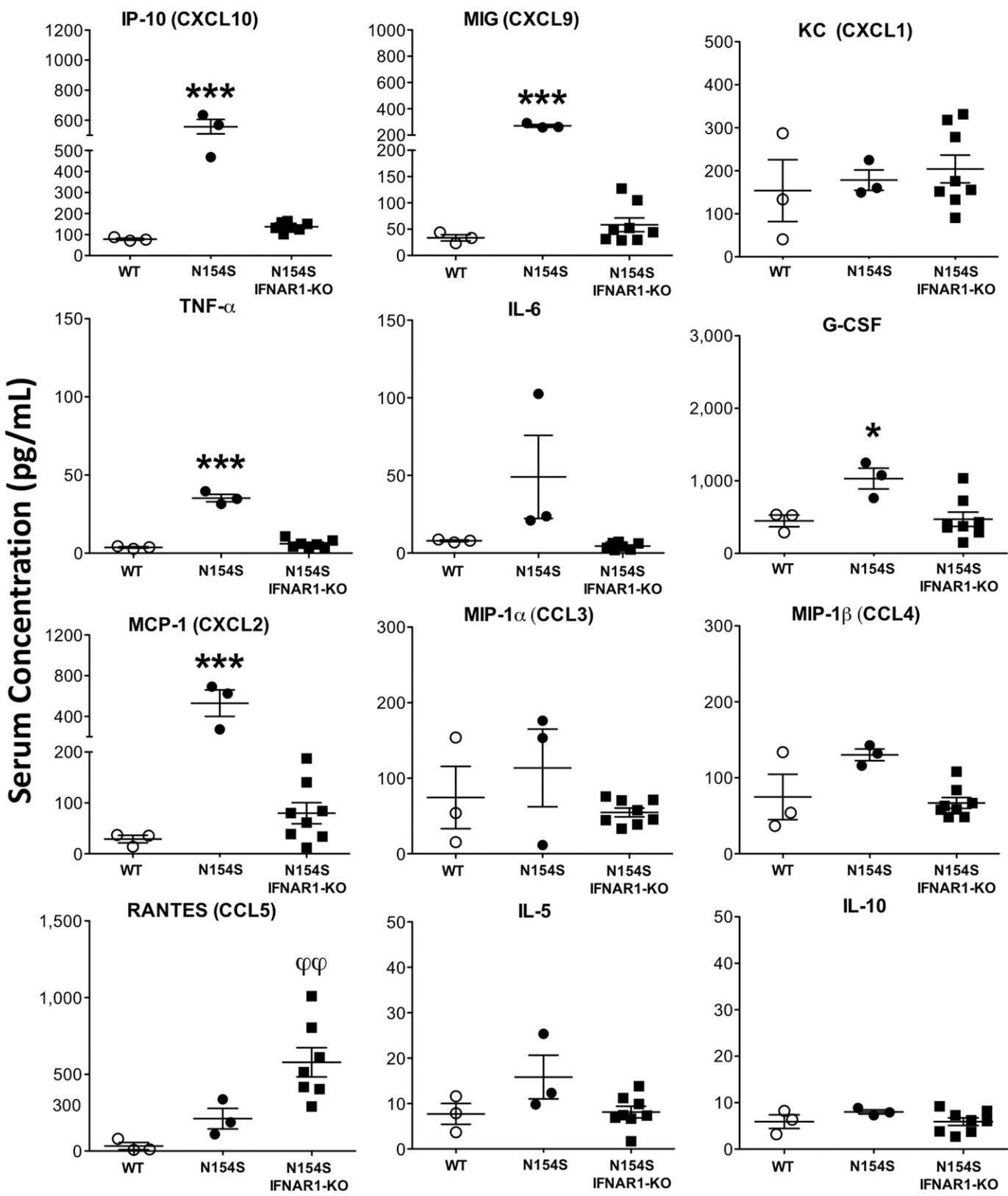

**Figure 9. Absence of systemic hyper-cytokinemia in *hSTING-N154S/Ifnar1*-KO mice.**
Serum cytokine levels (pg/ml) in age-matched WT, *hSTING-N154S* (N154S), and *hSTING-N154S/Ifnar1*-KO mice as measured by Luminex 31-plex murine cytokine array. Each symbol represents an individual mouse, and horizontal lines represent the mean ± SEM, n = 3 for WT and *hSTING-N154S*; n = 8 for *hSTING-N154S/Ifnar1*-KO. Statistical

## Serum cytokine assays

Blood was collected by cardiac puncture from deeply anesthetized 3–6-mo-old mice and transferred to 1.6-ml Eppendorf tubes. Co-agulated blood was centrifuged at 8,765 $g$ for 10 min at 4°C, and serum supernatant was collected and stored at −20°C until assayed. The samples were assayed using 31- and 13-plex Milliplex murine cytokine/chemokine arrays (Millipore) on a Luminex 200 system (Luminex Corp.) by Eve Technologies (Eve Technologies), and ELISA for mIFN-$\alpha$ (LumiKine Xpress; InvivoGen) and mIFN$\beta$ (LumiKine; InvivoGen). Results are presented in picograms per milliliter.

## XRM

Microfil$^R$ perfusion was performed as we previously described (Downey et al, 2012). High-resolution imaging of paws from mice perfused with the radio-opaque Microfil$^R$ was carried out with a Zeiss Xradia 520 Versa. XRM is distinct from the traditional microcomputed tomography as it combines both geometric mag-nification and optical objectives of microscopy to achieve higher spatial resolution at a relatively longer working distance (Sakdinawat & Attwood, 2010; Zhu et al, 2015). For XRM, front paws were sealed in centrifuge tubes containing neutral buffered saline. Both low-energy (40 kVp voltage, 3 W power) and high-energy (150 kVp voltage, 10 W power with a custom filter) XRM scans were performed on the same sample sequentially using the 0.4# ob-jective, which is sensitive to high-energy photons. To achieve a high signal-to-noise ratio, 2,501 projections were collected per rotation with each single projection exposure time of 3 s for low energy and 1.5 s for high energy. Obtaining the raw data sets required ~6 h of scanning time per paw.

## Hematopoietic cell isolation, BAL, and flow cytometry

Single-cell suspensions were prepared from lung tissue, peripheral blood, and BAL fluid from *hSTING-N154S* and control mice (n = 4–5 per group) as follows. Peripheral blood was collected in a 1-ml Eppendorf tube with 100 U/ml of heparin. Collected blood was lysed twice with RBC Lysis Buffer (BioLegend). A 100-$\mu$l aliquot of blood was lysed/fixed using a RBC Lysis/Fixation Solution (BioLegend), washed twice, and then counted by hemocytometer. For BAL, P90 polyethylene tubing (INSTECH) was inserted into the trachea at-tached with a 20-gauge 1-ml syringe. BAL was washed twice with 500 $\mu$l PBS supplemented with 2 mM EDTA. Before harvesting, the lungs were flushed with PBS via injection into the right ventricle. Lung tissues were then minced with scissors in ice-cold PBS and digested with collagenase IV (Worthingham; 80 U/ml) for 30–40 min at 37°C. After digestion, tissues were passed through a 70-$\mu$m filter and washed twice with FACS buffer (PBS supplemented with 2% FBS and 2 mM EDTA). BAL and lung preparations were lysed twice with RBC Lysis Buffer, washed twice with FACS buffer, and the cells were counted by hemocytometer. Lung, peripheral blood, and BAL single-cell suspensions were blocked using Fc-Block solution (Anti-mouse CD16/32; BioXCell) for 30 min on ice. The cells were then stained with APC-CD4 (Clone GK1.5), PE-CD8a (Clone 53.6-7), FITC-F4/80 (Clone BM8), FITC-Ly6G (Clone 1A8), BV605-CD19 (Clone 6D5), and PerCP/Cy5.5-CD11b (Clone M1/70) at 1:100 dilution in FACS Buffer for another 30 min on ice; all antibodies were purchased from BioLegend. Subsequently, the cells were washed and analyzed using a flow cytometer (BD FACS Canto), and flow cytometry data were analyzed using FlowJo software (version 10.2).

The thymus, spleen, and lymph nodes from *hSTING-N154S* mice, their WT littermates, and *mSting*-KO mice were examined for CD4$^+$ and CD8$^+$ cell populations. Harvested splenic tissue samples were mechanically disrupted to obtain single-cell suspensions and RBCs lysed using hemolysis buffer (ACK cell lysis buffer). The cells were then stained with anti-CD8-PerCP (clone 53-6.7; BD Pharmingen) and anti-CD4-FITC (clone GK1.5; BD Pharmingen). Flow cytometry data were acquired using a FACSCalibur flow cytometer (BD Bio-sciences) and analyzed using FlowJo software v8.6 (Tree Star). Leukocyte populations were selected using forward scatter/side scatter (FSC/SSC) and samples measured with a minimum of 10$^4$ counts.

## Intracellular staining of *hSTING* in hematopoietic cells

Single-cell suspensions were prepared from spleens and lungs of *hSTING-N154S*, WT control, and *mSting*g-KO mice (n = 6, n = 3, and n = 3, respectively). The lungs were flushed with 10 ml of saline through the right ventricle, then harvested and minced with surgical scissors, placed in 5 ml of PBS on ice, and then processed with the gentleMACS tissue dissociator. To further digest the tissue, the lung samples were incubated with dispase (2.5 U/ml) for 30 min at 37°C in a $CO_2$ in-cubator. The lung homogenate was then passed through a 100-$\mu$M filter into a 50-ml Falcon tube containing 10 ml PBS, centrifuged (650 $g$, 4°C, 5 min), and were treated with 1× RBC Lysis Buffer (BioLegend) for 3 min. For spleen, these tissues were placed in 1× PBS, ground between the rough sides of frosted glass slides, and then transferred to a 50-ml Falcon tube, centrifuged (234 $g$, 4°C, 5 min), and treated with hemolysis buffer (ACK cell lysis buffer) for 5 min at room temperature. The samples were then washed thrice with 1× PBS and incubated with anti-CD16/32 (FcBlock-BioXCell) for 30 min on ice. After washing the samples thrice with FACS wash (PBS, 2% FBS, and 0.002 M EDTA), the samples were transferred to a 96-well plate where antibody cocktails for cell surface markers were added to the cor-responding wells and stained on ice for 30 min. All antibodies were used at 1:200 dilutions. The cells from the spleen were stained with FITC-CD3, PerCPcy5.5-CD11b, and PE-CD19 and from the lung, with PE-CD31, and PE-cy7-CD45. After washing the samples thrice with FACS wash, the cells were fixed and permeabilized using the Foxp3 fixation/permeabilization working solution from eBioscience Foxp3/transcription factor staining buffer set (Invitrogen 00-5522-00). After

significance between data sets was assessed by one-way ANOVA followed by Tukey's multiple comparisons post hoc test between all groups. Significant differences between *hSTING-N154S* mice and WT or *hSTING-N154S/Ifnar1*-KO denoted by \*$P < 0.05$, \*\*$P < 0.01$, \*\*\*$P < 0.001$; WT and *hSTING*-N154S/*Ifnar1*-KO differences denoted by $\varphi$ $P < 0.05$. LIX, EOTAXIN, MIP-2, M-CSF, GM-CSF, IFN$\gamma$, VEGF, IL-1$\alpha$, IL-1$\beta$, IL-2, IL-3, IL-4, IL-7, IL-9, IL-12p40, IL-12p70, IL-13, IL-15, IL-17, and LIF were also measured; however, no differences were observed between the groups (data not shown).

1-h incubation, the cells were washed in 1× permeabilization buffer and incubated with Alexa Fluor 647–hSTING (1:40 dilution; BD pharminogen) for 30 min, they were washed and analyzed using a flow cytometer (BD FACS Canto). Flow cytometry data were analyzed using FlowJo software (version 10.2).

### Western blot analysis

To examine STING protein expression, splenic homogenates (10% wt/vol) from the various mouse phenotypes were prepared in extraction buffer (0.15 M NaCl, 5 mM EDTA, 1% Triton-X 100, and 10 mM Tris–HCl, pH 7.4) with the addition of a protease inhibitor cocktail (Complete, Roche Diagnostic GmbH). Protein concentrations were determined by Bradford assay. Proteins were separated on 12% SDS-polyacrylamide gels and transferred onto PVDF membranes. Membranes were blocked in 3% BSA/TBST and then incubated with an anti-STING rabbit polyclonal antibody (1:1,500 dilution; Cell Signaling Technology Inc.) that recognizes both human and mouse STING for 24 h at 4°C. As positive controls, two human colorectal cancer cell (CRC) lines known to express hSTING protein were used: HT29 and HCT116. An HRP-conjugated donkey antirabbit secondary antibody was used. STING protein bands were then visualized with SuperSignal West Pico chemiluminescence substrate and quantified using a calibrated imaging densitometer equipped with Quantity One software (Bio-Rad). The membranes were then stripped and re-probed with an anti-$\beta$-actin antibody as a loading control (Sigma-Aldrich). For spleen analyses, 40 $\mu$g of protein/lane and for CRC cell protein, 10 $\mu$g/lane were loaded.

### ANA detection

ANA testing used HEp-2 cell substrates (HEp-2000; Immuno-Concepts Inc.) to screen for mouse autoantibodies by indirect immunofluorescence (IIF) at a screening dilution of 1/160. All available samples were also tested for ANA specificities included in the ENA screening panel (chromatin, ribosomal P, Sm, U1RNP [ribonucleoprotein], SS-A/Ro60, Ro52/TRIM21, SS-B/La, Scl-70 [topoisomerase I], Jo-1 [histidyl tRNA synthetase]) by addressable laser bead immunoassay (FIDIS; TheraDiag), other myositis-related antibodies (OJ, TIF1γ, PL-12, SAE, EJ, MDA5, PL-7, SRP, NXP2, MI-2) by line immunoassay (Euroimmun GmbH), anti-centromere by IIF pattern on HEp-2 cells, and dsDNA by the *Crithidia lucilliae* IIF test (ImmunoConcepts). Antibodies to DFS70/LEDGF were detected by chemiluminescence immunoassay (QUANTA Flash DFS70; INOVA Diagnostics).

### Statistics

Statistical analysis was performed using GraphPad PRISM software (v5.0b). Variance between sample sets was estimated by a one-way ANOVA followed by Tukey's multiple comparisons post hoc test. Analysis of data from the qRT-PCR experiments was performed using a paired *t* test. Unpaired *t* tests were used to compare mean serum cytokine levels and leukocyte counts between *hSTING*-N154S mice and littermate controls. *P*-values of <0.05 were considered significant.

## Supplementary Information

## Acknowledgements

We are very grateful to Meifeng Zhang for performing the ANA determinations, the assistance of Wei Liu with the XRM, to Dragana Ponjevic for histological sectioning, to Elaine De Heuvel for her assistance with the histopathology preparation, to Cameron Fielding for generating the transgenic lines in the Transgenic Core Facility in the Clara Christie Centre for Mouse Genomics, and the Flow Cytometry Core Facility at the University of Calgary. K Henare was the recipient of a Postdoctoral Fellowship from the Health Research Council of New Zealand (Project Grant Number 15/446), administered by the University of Auckland. BG Yipp was supported by a Tier II Canada Research Chair in Pulmonary Immunology, Inflammation and Host Defense. The Zeiss Xradia 520 versa XRM instrument was obtained via a grant from the Canadian Foundation for Innovation. This study was supported by an Operating Grant from the Canadian Institutes for Health Research (to FR Jirik).

### Author Contributions

GR Martin: conceptualization, data curation, formal analysis, supervision, investigation, methodology, and writing—original draft, review, and editing.
K Henare: data curation, investigation, and methodology.
C Salazar-Arcila: data curation, formal analysis, investigation, and methodology.
T Scheidl-Yee: data curation, formal analysis, investigation, and methodology.
LJ Eggen: resources, data curation, investigation, and project administration.
PP Tailor: data curation, formal analysis, investigation, and methodology.
JH Kim: resources, data curation, formal analysis, investigation, and methodology.
J Podstawka: data curation, investigation, and methodology.
MJ Fritzler: resources, data curation, formal analysis, and investigation.
MM Kelly: data curation, formal analysis, and investigation.
BG Yipp: conceptualization, resources, data curation, formal analysis, supervision, funding acquisition, investigation, and writing—original draft, review, and editing.
FR Jirik: conceptualization, resources, supervision, funding acquisition, investigation, project administration, and writing—original draft, review, and editing.

### Conflict of Interest Statement

The authors declare that they have no conflict of interest.

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
