## [Reviewer comments · Life Science Alliance]

Life Science Alliance

A constitutively-active human STING mutant produces an Ifnar1-dependent vasculopathy

Gary Martin, Kimiora Henare, Carlolina Salazar-Arcila, Teresa Scheidl-Yee, Laura Eggen, Pankaj Tailor, Jung Kim, John Podstawka, Marvin Fritzler, Margaret Kelly, Bryan Yipp, and Frank Jirik

DOI: <https://doi.org/10.26508/lsa.201800215>

Corresponding author(s): Gary Martin, University of Calgary and Frank Jirik, University of Calgary

Review Timeline:

Submission Date:	2018-10-11
Editorial Decision:	2018-11-03
Revision Received:	2019-05-25
Editorial Decision:	2019-06-05
Revision Received:	2019-06-11
Accepted:	2019-06-12

Scientific Editor: Andrea Leibfried

Transaction Report:

November 3, 2018

Re: Life Science Alliance manuscript #LSA-2018-00215-T

Dr. Gary R Martin
University of Calgary
Department of Biochemistry and Molecular Biology
3A25 HRIC
3280 Hospital Drive NW
Calgary, AB T2N 4Z6
Canada

Dear Dr. Martin,

Thank you for submitting your manuscript entitled "Expression of a constitutively-active human STING mutant in hematopoietic cells produces an Ifnar1-dependent vasculopathy in mice." to Life Science Alliance. The manuscript was assessed by expert reviewers, whose comments are appended to this letter.

As you will see, the reviewers appreciate your analyses and the mouse model provided, and they make good suggestions on how to strengthen your manuscript to allow publication upon minor revision. In our view, most points can get addressed by re-writing/acknowledging drawbacks or lack of insight and by changing the data representation. Importantly however, the request for STING expression level analysis as a control (reviewer #1, point 1) should get addressed, and the revision should acknowledge that the experimental set-up could have been better (reviewer #1 and #3).

Thank you for this interesting contribution to Life Science Alliance. We are looking forward to

receiving your revised manuscript.

Sincerely,

- A letter addressing the reviewers' comments point by point.
- An editable version of the final text (.DOC or .DOCX) is needed for copyediting (no PDFs).
- High-resolution figure, supplementary figure and video files uploaded as individual files: See our detailed guidelines for preparing your production-ready images, <http://life-science-alliance.org/authorguide>
- Summary blurb (enter in submission system): A short text summarizing in a single sentence the study (max. 200 characters including spaces). This text is used in conjunction with the titles of papers, hence should be informative and complementary to the title and running title. It should describe the context and significance of the findings for a general readership; it should be written in the present tense and refer to the work in the third person. Author names should not be mentioned.

B. MANUSCRIPT ORGANIZATION AND FORMATTING:

Full guidelines are available on our Instructions for Authors page, <http://life-science-alliance.org/authorguide>

Reviewer #1 (Comments to the Authors (Required)):

In this work the authors have characterized a TG mouse strain harbouring the hSTING N154S mutation known to cause SAVI. Based on the data presented, the authors conclude that disease development in the mice shows some similarities with human SAVI, and that disease development is dependent on type I IFN. Although the work is potentially interesting, and adds to an expanding literature on modeling of SAVI in mouse models, the work has flaws that need to be addressed.

1. The mouse is made as a KI from a STING KO mouse. Therefore, the correct control would be a KI of WT STING. This represents a significant problem, and should be presented to the reader. To allow interpretation of the data, the authors should as a minimum compare STING expression in WT versus TG mice.
2. Along the same lines, the fact that STING is expressed from a Vav1 promoter is not well motivated, and further undermines WT and STING KO mice as the correct controls. It is important that this is described very clearly in the main text.
3. The observed acral necrosis is interesting. The authors should test what type of cell death that is occurring (apoptosis, necroptosis, pyroptosis....)

Reviewer #2 (Comments to the Authors (Required)):

Martin et al report on the study of a new mouse model expressing a hSTING-N154S transgene under the vav promoter, thereby restricting its expression in hematopoietic tissues. Interestingly, these animals developed an inflammatory disease similar to SAVI patients with the exception of a lung disease. Crossing these mice to an ifnar-1 ko background fully reversed the phenotype and prevented the development of the disease.

This is a very interesting (Sting) animal model partially mimicking the human pathology. Very strikingly, this model is interferon dependent, whereas previously described knock-in mice expressing an activating mSting mutation are independent of the interferon pathway.

Whereas the authors discussed the phenotype variability between the transgenic and the knock-in models with regards to clinical symptoms, they do not discuss the interferon dependence of their model. A paragraph discussing the restricted expression of the hSTING-transgene in the hematopoietic compartment versus an ubiquitous expression of the mSting should be provided. Another striking observation is the production of auto-antibodies, a feature not seen in knock-in mice. However, there is very few B cells in the transgenic mice. How the authors could explain this apparent discrepancy?

Reviewer #3 (Comments to the Authors (Required)):

Uncontrolled activation of the innate immune system has been linked to systemic autoimmunity. In this regard, investigating conditions arising due to chronic activation of the cGAS-STING pathway have been very informative. Here, Martin et al. report the generation of a transgenic mouse model that expresses a mutant version of human STING (hSTING-N154S), which has been shown to be constitutively active. This particular mutation causes a rare monogenic autosomal dominant autoinflammatory disease called STING-associated vasculopathy with onset in infancy (SAVI). In the transgenic mouse model expression of mutant hSTING-N154S is controlled by the Vav1 regulatory element, which is highly active in hematopoietic cells. Founder mice that expressed the

transgene recapitulated important features of SAVI such as reduced weight, lymphopenia, inflammation of the skin, myositis and vasculitis. Furthermore, the authors detect elevated expression of various pro-inflammatory cytokines and type I interferons (IFN), as well as increased titers of ANAs in hSTING-N154S mice compared with controls. However, the mouse model fails to reproduce severe inflammatory lung disease, a hallmark of SAVI. Most strikingly, pathology and cytokine expression in hSTING-N154S could be rescued by genetic ablation of the type I IFN receptor (*Ifnar1*).

Although two SAVI mouse models have already been published, the model that Martin et al., present is highly interesting and well worth publishing. It differs conceptually from the two recently published SAVI mouse models, both of which modified the endogenous STING-encoding *Tmem173* gene to carry the mutation homologous to SAVI-mutations in human STING. The observation that *Ifnar1*-deficiency rescues hSTING-N154S mice is puzzling since in the two knock in mouse models inactivation of type I IFN signaling had only modest effects on the pathology. Characterization of the newly generated SAVI model has been carried out thoroughly and data are presented clearly in the figures. However, some aspects should be investigated in more depth to support the main conclusion drawn here before the manuscript can be accepted for publication.

Major concerns:

1. In the present model expression of hSTING-N154S is driven by the *Vav1*-regulatory sequence, which according to the authors confines expression of the transgene to the hematopoietic lineage. The *Vav* -promoter appears to be switched on also in endothelial cells (Georgiades et al., *Genesis* 2002). Albeit being only transiently activated this would cause the variant hSTING to be expressed in endothelial cells, which might initiate a pathogenic chain that is later aggravated by hSTING expressing hematopoietic cells. Given the prominent vascular phenotype of the hSTING-N154S mice reported here and that the authors conclude expression of the mutant hSTING in hematopoietic cells causes vasculitis, it is vital to demonstrate that expression of the variant is entirely restricted to the hematopoietic compartment. If this is not the case a more careful interpretation of most data and revision of the manuscripts title is required.
2. It is not sufficient to demonstrate expression of the transgene only in splenic B cells. The authors need to provide evidence for expression in various hematopoietic cells. Why do only 35% of splenic B-cells express hSTING? How does this compare to the expression level of endogenous murine STING? This could be addressed by co-staining mSTING and hSTING in the same cells with individual fluorochromes. The protocol for staining hSTING is inconclusive, as detection of intracellular STING requires a permeabilization step and not only "co-incubation" of splenocytes with anti-CD19 and anti-hSTING antibodies. Can the authors please provide an exact protocol on how the staining and the gating was performed?
3. In contrast to the two previously reported mouse models of SAVI, deletion of *Ifnar1* completely rescues pathology in hSTING-N154S mice. This is an important but also highly puzzling observation, which needs to be discussed in more detail. Can the authors comment on what might be the difference in their model compared with the knock in models? How does the expression level of their transgene compare to endogenous mSTING (see comment 1&2)? Does hSTING fail to activate the murine *NfκB* pathway and therefore observed expression of TNF and IL-6 are not cell-intrinsically activated as in the other SAVI models, but rather a secondary response to IFN-induced inflammation and tissue damage? Analyzing the cytokine expression pattern (e.g *NfκB* targets) of primary MEFs from hSTING-N154S mice, which are not exposed to exogenous inflammatory stimuli could potentially shed light on this question? Does hSTING-N154S form active heterodimers with endogenous mSTING - this could be addressed by immunofluorescence?

4. Heterozygous expression of mSTING-N154S from the endogenous Tmem173 locus resulted in a reduced lifespan and altered Mendelian ratios (Warner et al., 2017). Warner and colleagues also reported frequent abortions. Does expression of hSTING-N154S affect survival of transgene-positive mice? Does transmission of the transgene fulfill Mendelian ratios in both genders?

Minor points:

5. Figure 1A/B: The main text refers to three founder mice while the graphs show more than three data points. Please indicate that the offspring of the three different founders are displayed? What is the "n" per group and how many mice from which founder were analyzed? Is the difference statistically significant?

6. Also regarding the three founders that developed a phenotype: Offspring of which founder have been used in all subsequent experiments? This is of interest because there seem to be significant differences in the severity of the phenotype between the founders as shown in Figure S1.

7. Mice were generated on a mixed C57BL/6 x DBA background and not on a pure C57BL/6. How many times have they been backcrossed for the study? Inconsistent genetic backgrounds might contribute to observations different from previously published findings.

8. Please provide quantifications of all histopathological observations, e.g. scores.

9. Figure S1: No IFN α levels are shown, but they are mentioned in the title of the figure. Can the authors include these interesting data? 1504 & 1501 likely refer to different founders? If so, what is the paw thickness of the third founder?

10. Figure 7&8: More mice hSTING-N154S Ifnar1 $^{-/-}$ mice should be analyzed in the rescue experiments.

11. Figure S2: Histology of the respective controls should be shown to allow a broad range of readers to immediately identify aberrant effects in hSTING-N154S mice.

12. Figure 2: What is the age of the mice? Is there inflammation in lungs of aged mice? Do the rare inflammatory foci expand?

13. Figure 4: Can the authors provide data on the cellularity of the spleen, thymus and lymph nodes? Can the authors provide absolute numbers, as these are more informative and would probably reveal a more drastic effect of transgene expression on the lymphoid compartment? Why have B cells and myeloid cells not been analyzed in these organs?

The legend says group size >10, but STING KO is n=4, please correct.

Figure 4 D&E: Please only provide the total number of mice per group that has been analyzed in five experiments.

14. Figure 5: The legend says n=9 but number of symbols does not match. The legend also refers to Figure S2 but the linked dataset is represented in Figure S3. Please also harmonize the group sizes in Figures 5 and S3, as they appear to show analysis of the same mice. Also indicate how many independent experiments have been carried out.

15. I would suggest to combine all data on lymphopenia in one section and to not revisit this topic

after discussion of different topics. My suggestion would be to reorganize the manuscript into fewer defined sections addressing the macroscopic phenotype, hematopoietic parameters, cytokine expression, autoimmunity and the role of Ifnar signaling in the pathology.

16. Figure 6B - left: What is the CD19intermediate/Ly6G-positive population? How were the gating and/or compensation for the staining performed?

17. Figure S4: Based on which criteria has the gate been set? What is the CD11b high F4/80 low population that is only present in the WT but almost absent in the hSTING-N154S lungs?

18. Figure S5: At which age have the mice been analyzed? Can the authors provide the full dataset of the immune-fluorescence pattern of the ANAs and on the auto-antigens that they tested, including titers for the reactivity against the auto-antigens? Are there ANAs to human STING in these mice? Can the authors provide any statistics on whether the ANA frequency really differs between the groups?

19. Figure 7: Are littermate controls shown? Do all mice have the same gender?
In 7D, is the difference in body weight between WT and hSTING-N153S not statistically significant? Lines above the groups that have been compared could alternatively label comparisons, which might provide a better instant understanding of the graphs.

Reviewer #1 (Comments to the Authors (Required)):

In this work the authors have characterized a TG mouse strain harbouring the hSTING N154S mutation known to cause SAVI. Based on the data presented, the authors conclude that disease development in the mice shows some similarities with human SAVI, and that disease development is dependent on type I IFN. Although the work is potentially interesting, and adds to an expanding literature on modeling of SAVI in mouse models, the work has flaws that needs to be addressed.

1. The mouse is made as a KI from a STING KO mouse. Therefore, the correct control would be a KI of WT STING. This represents a significant problem, and should be presented to the reader. To allow interpretation of the data, the authors should at a minimum compare STING expression in WT versus TG mice.

Response: The question is based on an error made by the reviewer, namely, the various independent transgenic founder lines that were generated represent the products of random chromosomal integrations of the Vav1-hSTING-N154S transgene; they are not knock-ins, nor are they on an mSting KO background. In most of experimental groups, both endogenous WT mSting alleles are still present along with the Vav1 transgene.

While Vav1-WT hSTING transgenics would be a useful control, in view of our finding that the phenotype our Vav1-mutant hSTING model differs significantly from the various mSting KIs already published, a more pertinent control would be to express the cognate mSting mutant under the control of the Vav1 gene promoter. If this replicated the phenotype of the Vav1-mutant hSTING, then it would argue that the phenotypic discordance between our transgenics and the various knock-ins was a reflection of altered expression pattern stemming from the use of the Vav1-gene promoter, rather than the endogenous murine Sting gene promoter. The generation and characterization of any new transgenic lines could take up to one year, and hence would need to be the subject of a follow-up manuscript.

Regarding comparison of expression levels between the endogenous mSting and the transgene-derived mutant hSTING, it is correct that this is an important point: We examined the splenic expression levels of STING in WT and TG+ by Western blot analyses and by FACS. We found that there was no significant elevation of total STING expression when mice that expressed the Vav1-hSTING-N154S (N154S) transgene were compared to WT mice. In fact there were relatively low levels of transgene-derived mutant STING expression observed in splenic lysates when Vav1-hSTING-N154S mice were crossed onto an mSting^{Gt/Gt} KO background and probed with an antibody that detects both human and murine STING.

Furthermore, we analyzed various splenic populations, including CD3⁺ (T cells), CD11b⁺ (macrophages) and CD19⁺ (B cells), as well as CD31⁺ endothelial cells that were isolated from the lung, to examine Vav1-mutant hSTING expression using a human specific hSTING directly fluor-conjugated antibody. We found that only the transgene-positive cells expressed the mutant hSTING. Examining endothelial cells, since there was a possibility that the Vav-1 promoter may have led to ectopic hSTING expression in this non-hematopoietic population, we did not detect hSTING expression.

These various results have now been incorporated into the revised manuscript both as figures and in the text.

2. Along the same lines, the fact that STING is expressed from a Vav1 promoter is not well motivated, and further undermines WT and STING KO mice as the correct controls. It is important that this is described very clearly in the main text.

Response: Our hypothesis was that the SAVI-like interferonopathy phenotype was likely mediated by hematopoietic cells, since many of these cells (e.g. NK cells, plasmacytoid DCs, macrophages) are potent Type I interferon generators. For this reason, we wanted to directly assess the potential role of hematopoietic cell expression of the hSTING-N154S mutant. The last paragraph in the introduction now includes the following statements:

“Murine models for SAVI will facilitate studies of the pathogenesis of this genetic disease and allow for the evaluation of new therapies. Since specific cells of hematopoietic origin are well established as key type 1 IFN and proinflammatory cytokine producers, we hypothesized that these cells were likely involved in the pathogenesis of SAVI. To examine this possibility, we generated transgenic mice whereby expression of a SAVI-associated *hSTING* mutation (N154S) was targeted to hematopoietic cells by the *Vav1* gene promoter...”

Fortunately, using our Vav1 transgenic approach, we were able to generate a model whose phenotype shows several of the key features of human SAVI. However, we point out that our model lacks any significant lung inflammation, thus raising the possibility that the lung inflammation requires lung parenchymal cell expression of the mutant STING. This is an important finding in its own right.

Furthermore, anticipating that future use of our model in preclinical drug assessments aimed at testing novel STING inhibitors eventually destined for human use, that it would be advantageous to have a disease model that relied on the activity of the human STING protein. Especially since there are sequence differences between the murine and human STING proteins.

Addressing the issue of other transgenic controls, as addressed in #1 above, and lies outside of the scope of this manuscript.

3. The observed acral necrosis is interesting. The authors should test what type of cell death that is occurring (apoptosis, necroptosis, pyroptosis....)

Response: The X-ray microscopy imaging, as well as the histology showing vessels occlusions, and the finding of large areas of necrosis in the paws, as well as tail and ear damage are entirely consistent with ischemic necrosis due to arteriolar thrombosis (Figures have already been presented of these phenomena that occur with full penetrance in all the transgenic mice). We have examined the affected areas by TUNEL, but found no widespread evidence of apoptosis. However, in some rare areas that contain abundant infiltrating hematopoietic cells, several TUNEL + cells were present. This is likely consistent with lymphocyte apoptosis that has been well described in response to STING activation.

Reviewer #2 (Comments to the Authors (Required)):

Martin et al report on the study of a new mouse model expressing a hSTING-N154S transgene under the vav promoter, thereby restricting its expression in hematopoietic tissues. Interestingly, these animals developed an inflammatory disease similar to SAVI patients with the exception of a lung disease. Crossing these mice to an ifnar-1 ko background fully reversed the phenotype and prevented the development of the disease.

This is a very interesting (Sting) animal model partially mimicking the human pathology. Very strikingly, this model is interferon dependent, whereas previously described knock-in mice expressing an activating mSting mutation are independent of the interferon pathway.

Whereas the authors discussed the phenotype variability between the transgenic and the knock-in models with regards to clinical symptoms, they do not discuss the interferon dependence of their model. A paragraph discussing the restricted expression of the hSTING-transgene in the hematopoietic compartment versus an ubiquitous expression of the mSting should be provided.

Response: As correctly suggested by the reviewer, it is plausible that cell type-specific expression levels in the various hematopoietic populations has a role explaining the phenotypic differences between the mSting mutant knock-ins and our transgenics, given that Vav1 gene promoter activity in the transgene may differ (qualitatively and/or quantitatively) from that of the endogenous mSting locus that drove expression of the mSting KI mutants. In the discussion, we indicate that it is possible that circulating levels of type I IFNs and other cytokines may be higher in our mice than in the mSting KI models, and this could potentially promote the development of the vasculopathy.

For example, as stated in the Discussion:

“Regarding lung involvement, Warner *et al.* described the phenotype of mice having an N153S knock-in mutation of *mSting*. These mice developed severe lung inflammation, skin ulceration, as well as hyper-cytokemia and lymphopenia. (Warner et al., 2017) Similar to the *hSTING*-N154S mice, 4 to 6 month old *mSting*-N153S mice had elevated serum pro-inflammatory mediators, (Warner et al., 2017) albeit at lower levels than those seen in *hSTING*-N154S mice.”

We now elaborate further on the differences between our model and the mSting KIs in the Discussion, as follows:

“In view of the phenotypic differences between our mutant *hSTING* transgenics and the *mSting* knock-ins, an important question remains: why is the phenotype of our transgenic model different from that of the various *mSting* knock-in transgenics that were not reported to develop acral necrosis? One obvious possibility concerns the use of an ectopic gene promoter to drive *hSTING*-N154S expression. The *Vav1*-gene promoter is unlikely to be subject to the same regulation as the endogenous *mSting* gene promoter, a factor that may have led to mutant hSTING protein expression being higher than mutant mSting expression. However, when *Vav1-hSTING*-N154S mice were placed on an *mSting* KO background, relatively low levels of STING expression were seen using an antibody that detect both human and murine STING. A comparison of protein expression levels between our mutant *hSTING* mice and the biallelic mutant *mSting* knock-ins would be of interest, since the constitutively-active mutant *hSTING* protein likely undergoes rapid degradation.”

Regarding the restricted expression versus the ubiquitous mSting expression we had added the following to the Discussion (also please see response #2 to Reviewer 1):

“Why do the *Vav1-hSTING*-N154S transgenics invariably develop prominent paw inflammation and acral necrosis? It has been reported that the *Vav1* gene promoter may be expressed in endothelial cells (Joseph et al., 2013) and thus expression in these cells may have been a factor in the vascular pathology. Although it is possible that mutant *hSTING* protein expression levels were below the sensitivity of the intracellular detection method we used, *hSTING* expression was not detected in the endothelial cells that had been isolated from the lung. However, if the acral vasculopathy was indeed dependent on mutant *hSTING* expression in endothelium, why is it that *mSting* mutant knock-ins did not develop this pathology, given that *mSting* is thought to be ubiquitously-expressed?....”

Furthermore, cell type-specific expression can be an important variable when it comes to phenotypic expression, as exemplified by the lack of a lung phenotype in our transgenics. Unlike humans with activating *STING* mutations, and the *mSting* mutant knock-ins, our *hSTING*-N154S mice did not develop significant lung inflammation. This was also consistent with our findings that lungs and bronchoalveolar lavage fluid of *hSTING*-N154S mice did not exhibit significant increases in inflammatory cell numbers, except for neutrophils (consistent with the neutrophilia). Since *Vav1-hSTING*-N154S expression was largely confined to hematopoietic cells, the lack of lung disease leads to the hypothesis that expression of constitutively-active STING protein in lung parenchymal cells are required for the lung inflammation to occur.

Another striking observations is the production of auto-antibodies, a feature not seen in knock-in mice. However, there is very few B cells in the transgenic mice. How the authors could explain this apparent discrepancy?

Response: We have no simple explanation for the presence of autoantibodies in our transgenics, however, we should point out are using the very sensitive and specific assays that

are carried out in the lab of Dr. M. Fritzler, a world expert when it comes to autoantibodies and rheumatological autoimmune diseases in general. The sensitivity of the assays is illustrated by the finding of positive ANAs, albeit tending to show lower titers, even in littermate WT B6 control animals. To further document this, we have now included a table containing the raw data for both the control and transgenic mice that includes ANA titers, immunofluorescence patterns and reactivities against a panel of potential autoantigens.

While it is correct that there are B cell and T cell depletions in all of the mutant Sting models (ours and the KIs), the disease phenotype in our transgenic mice evolve over many weeks, making it feasible that Th-dependent autoantibody-producing cells are generated early on such that long-lived plasma cells are established (and these might possibly be resistant to activated STING, or potentially, that the Vav1 promoter is much less active in these cells). Perhaps the time course of the disease is delayed in our transgenics (as compared to the Sting KIs). Also, the autoantibodies could result from Ig isotypes with long half-lives if autoimmunity developed at an early stage of disease evolution. We are not aware of whether there may be subsets of B cells that may be relatively resistant to death induction by activated STING (e.g. are B1 cells, or are differentiating B2 cells, or plasmablasts also affected). Can DCs with activated Sting, function as super-APCs?

Lastly, it is possible that B cell death is more profound in the mutant mSting knock-in models than in our hSTING mutant transgenics. Perhaps, sufficient B cells are still available in the latter mice to allow for autoantibody generation as is evident in our mice and in SAVI. The Vav1 gene promoter, after all, is not the same as the endogenous mSting promoter and hence this difference could be a factor affecting the progress of the disease, such that evidence of autoimmunity can develop.

In summary, this is clearly an interesting question that has been raised by the reviewer, and one that would merit a side by side comparison, for example, of the severity and temporal aspects of immune cell killing, in the mSting KI models versus our transgenics.

Reviewer #3 (Comments to the Authors (Required)):

Uncontrolled activation of the innate immune system has been linked to systemic autoimmunity. In this regard, investigating conditions arising due to chronic activation of the cGAS-STING pathway have been very informative. Here, Martin et al. report the generation of a transgenic mouse model that expresses a mutant version of human STING (hSTING-N154S), which has been shown to be constitutively active. This particular mutation causes a rare monogenic autosomal dominant autoinflammatory disease called STING-associated vasculopathy with onset in infancy (SAVI). In the transgenic mouse model expression of mutant hSTING-N154S is controlled by the Vav1 regulatory element, which is highly active in hematopoietic cells. Founder mice that expressed the transgene recapitulated important features of SAVI such as reduced weight, lymphopenia, inflammation of the skin, myositis and vasculitis. Furthermore, the authors detect elevated expression of various pro-inflammatory cytokines and type I interferons (IFN), as well as increased titers of ANAs in hSTING-N154S mice compared with controls. However, the mouse model fails to reproduce severe inflammatory lung disease, a hallmark of SAVI.

Most strikingly, pathology and cytokine expression in hSTING-N154S could be rescued by genetic ablation of the type I IFN receptor (Ifnar1).

Although two SAVI mouse models have already been published, the model that Martin et al., present is highly interesting and well worth publishing. It differs conceptually from the two recently published SAVI mouse models, both of which modified the endogenous STING-encoding Tmem173 gene to carry the mutation homologous to SAVI-mutations in human STING. The observation that Ifnar1-deficiency rescues hSTING-N154S mice is puzzling since in the two knock in mouse models inactivation of type I IFN signaling had only modest effects on the pathology. Characterization of the newly generated SAVI model has been carried out thoroughly and data are presented clearly in the figures. However, some aspects should be investigated in more depth to support the main conclusion drawn here before the manuscript can be accepted for publication.

Major concerns:

1. In the present model expression of hSTING-N154S is driven by the Vav1-regulatory sequence, which according to the authors confines expression of the transgene to the hematopoietic lineage. The Vav -promoter appears to be switched on also in endothelial cells (Georgiades et al., Genesis 2002). Albeit being only transiently activated this would cause the variant hSTING to be expressed in endothelial cells, which might initiate a pathogenic chain that is later aggravated by hSTING expressing hematopoietic cells. Given the prominent vascular phenotype of the hSTING-N154S mice reported here and that the authors conclude expression of the mutant hSTING in hematopoietic cells causes vasculitis, it is vital to demonstrate that expression of the variant is entirely restricted to the hematopoietic compartment. If this is not the case a more careful interpretation of most data and revision of the manuscripts title is required.

Response: It is correct that there is a possibility that the Vav1-gene promoter could have led to hSTING expression in endothelial cells, for this reason we isolated endothelial cells (CD31+ CD41-) from the lungs to determine whether hSTING expression was present, as detected by intracellular staining using a human-specific STING direct-conjugated monoclonal. TG+ splenic cells expressed hSTING, however, hSTING expression was undetectable in the endothelial cell population. However, this method, one that was suggested by the reviewer, does not exclude the possibility that low-level mutant hSTING protein expression in endothelium, which is below the ability of the intracellular staining technique to detect, is present in endothelial cells. These results and caveats are also now mentioned in the text. Since endogenous mSting is believed to be ubiquitously expressed, one would expect that the activated mSting protein, expressed by the mSting mutant knock-ins, would also be expressed in endothelium. Since acral necrosis was not reported in the knock-ins, this would argue against the possibility that low level mutant hSTING expression in endothelium plays a role in the vasculopathy phenotype that we observe. These points have been added to the discussion. Also see response #2 below.

2. It is not sufficient to demonstrate expression of the transgene only in splenic B cells. The authors need to provide evidence for expression in various hematopoietic cells. Why do only 35% of splenic B-cells express hSTING? How does this compare to the expression level of endogenous murine STING? This could be addressed by co-staining mSTING and

hSTING in the same cells with individual fluorochromes. The protocol for staining hSTING is inconclusive, as detection of intracellular STING requires a permeabilization step and not only "co-incubation" of splenocytes with anti-CD19 and anti-hSTING antibodies. Can the authors please provide an exact protocol on how the staining and the gating was performed?

Response: We agree with the reviewer's concern, and that problematic figure has been deleted. Thus, to address this we now show hSTING expression in the various hematopoietic cell types. Thus, we analyzed various splenic populations, including CD3⁺ (T cells), CD11b⁺ (macrophages) and CD19⁺ (B cells) to identify hSTING expression using a specific hSTING conjugated antibody. Only the TG⁺ cells expressed the mutant hSTING protein. Again, it is possible that each of the hSTING-negative cell populations do in fact express low levels of the mutant hSTING protein, below the level of detection of the intracellular staining technique. The splenic Westerns also support this possibility, since it is clear that the active mutant hSTING is not highly expressed when compared to the endogenous mSting.

We have introduced the new data alluded to above in the Results, as follows:

“hSTING-N154S protein expression in splenic cell populations and lysates

To examine mutant hSTING protein expression in the various splenic populations, including CD3⁺ (T cells), CD11b⁺ (macrophages) and CD19⁺ (B cells), as well as CD31⁺ endothelial isolated from the lung, we used a human-specific directly fluor-conjugated anti-hSTING antibody (**Fig S2**). **(A)** Representative dot plots show gating and relative proportions of CD3⁺ T cells in single cell suspensions of dissociated spleens obtained from *mSting* KO, WT, and *Vav1-hSTING-N154S* (SAVI) mice analyzed by multi-color flow cytometry. **(B)** Representative dot plots show the percentages of CD3⁺ hSTING⁺ cells from the spleens of *mSting* KO, WT, and *Vav1-hSTING-N154S* mice. Numbers below each gate are the percentage of cells within the corresponding gate. When hSTING-N154S protein expression was assessed in the various splenic cell populations, for example CD3⁺ (T cells), CD11b⁺ (macrophages) and CD19⁺ (B cells), only the transgene-positive cells expressed the mutant hSTING (**Fig 4C**). Percentages of positive cells in the transgenic spleens were relatively low possibly as a result of: (i) technical factors associated with the efficiency of the intracellular staining procedure in different cell types; (ii) expression levels per cell often being below the detection threshold; and (iii) the possibility of variegated transgene expression. To examine the possibility that the *Vav1* gene promoter may have resulted in expression of mutant hSTING protein in endothelial cells, we isolated CD31⁺ endothelial cells from the lung, a microvascular-rich organ. Using the intracellular staining we did not detect the mutant hSTING in this cell population (**Fig 4C and Fig S2A**). Immunoblotting with a polyclonal anti-STING antibody that recognized both human and mouse STING was then used to evaluate splenic expression of mSting in WT controls, in splenic lysates of *hSTING-154S* mice and *hSTING-N154S* mice on an *mSting* KO background. We found no significant increases in overall mSting and hSTING expression in the mice positive for the *hSTING-N154S* transgene as compared to mSting protein expression levels in WT mice (**Fig. 3, D**). In fact, there were relatively low levels of transgene-derived hSTING-N154S expression, best illustrated when *Vav1-hSTING-N154S* mice were placed onto an *mSting* KO background. As expected, mSting protein expression was absent in the spleens of *mSting* KO mice.”

3. In contrast to the two previously reported mouse models of SAVI, deletion of *Ifnar1* completely rescues pathology in *hSTING-N154S* mice. This is an important but also highly puzzling observation, which needs to be discussed in more detail. Can the authors comment on what might be the difference in their model compared with the knock in models? How does the expression level of their transgene compare to endogenous *mSTING* (see comment 1&2)? Does *hSTING* fail to activate the murine *NfκB* pathway and therefore observed expression of *TNF* and *IL-6* are not cell-intrinsically activated as in the other SAVI models, but rather a secondary response to IFN-induced inflammation and tissue damage? Analyzing the cytokine expression pattern (e.g *NfκB* targets) of primary MEFs from *hSTING-N154S* mice, which are not exposed to exogenous inflammatory stimuli could potentially shed light on this question? Does *hSTING-N154S* form active heterodimers with endogenous *mSTING* - this could be addressed by immunofluorescence?

Response: With regards to the expression levels of the transgenic *hSTING*, this was addressed earlier in our response and in the response to Reviewer #1. Regarding the question as to *hSTING*'s ability to activate the murine *NfκB* pathway, there is a possibility that the *TNF-α* and *IL-6* levels are increased in response to inflammation/tissue damage (ischemia and necrosis), however, we did not examine this or the cytokine expression pattern from unstimulated primary MEF cells derived from *hSTING* mice.

4. Heterozygous expression of *mSTING-N154S* from the endogenous *Tmem173* locus resulted in a reduced lifespan and altered Mendelian ratios (Warner et al., 2017). Warner and colleagues also reported frequent aborts. Does expression of *hSTING-N154S* affect survival of transgene-positive mice? Does transmission of the transgene fulfill Mendelian ratios in both genders?

Response: While our mice would have a reduced lifespan as a result of complications associated with the disease, the overall survival could not be completely assessed in our *hSTING* mice. We sacrificed at differing time points as, though the disease was penetrant, its time course and time to endpoint (sacrifice due to severity of disease) was variable. None of our mice were allowed to go beyond a given endpoint. The disease followed Mendelian ratios and was fully penetrant in mice of both sexes. We now mention this point in the discussion.

Minor points:

5. Figure 1A/B: The main text refers to three founder mice while the graphs show more than three data points. Please indicate that the offspring of the three different founders are displayed? What is the "n" per group and how many mice from which founder were analyzed? Is the difference statistically significant?

Response: Figure 1 A/B is the body weight changes of the *N154S* vs. WT mice (males and females) over time. The data plotted is from the 1501 founder line only. To add clarity, we

have added this information to the figure legend and in the methods/results sections. For example:

“By 8 - 10 wks of age, three of the five *hSTING*-N154S founder lines exhibited growth impairment, a failure to gain weight, and a reduced lifespan as a result of complications associated with the disease. (Fig. 1, A and B). However, overall survival could not absolutely be determined in our *hSTING*-154S mice as the time to endpoint (e.g. requiring sacrifice due to the severity of disease) was somewhat variable. We also observed that the disease in these three lines was prevalent and penetrant in males and females equally. To reduce variability, we selected the 1501 line (the most severe phenotype) and herein, all experimental observations reported are centered upon this line only.”

6. Also regarding the three founders that developed a phenotype: Offspring of which founder have been used in all subsequent experiments? This is of interest because there seem to be significant differences in the severity of the phenotype between the founders as shown in Figure S1.

Response: For the experiments in the main manuscript, the 1501 line was used as these displayed the more severe phenotype (for example, the paw swelling as detailed in Fig. S1). Please see the response to minor point #5.

7. Mice were generated on a mixed C57BL/6 x DBA background and not on a pure C57BL/6. How many times have they been backcrossed for the study? Inconsistent genetic backgrounds might contribute to observations different from previously published findings.

Response: This detail was described in the methods section (*Generation of hSTING-N154S transgenic lines*): “Five *hSTING*-N154S founders were identified and three of these that showed paw swelling, had been backcrossed onto the C57BL/6 background for between 5 and 8 generations during these the experiments”. We have added in information clarifying that the 1501 line was used in the experiments.

8. Please provide quantifications of all histopathological observations, e.g. scores.

Response: This is an unrealistic request. There is no formal scoring system that is available for the extent of acral (tail and ear) necrosis, myositis, cell infiltration density, or for the paw vasculopathy and areas of necrosis. All of these features are seen in every transgenic animal. The disease occurs in 100% of the mice and severity of pathology also worsens with age until the animals require euthanization. Unaffected animals are normal and thus the difference between experimental and control mice is black and white. The images provided are entirely representative.

9. Figure S1: No IFN α levels are shown, but they are mentioned in the title of the figure. Can the authors include these interesting data? 1504 & 1501 likely refer to different founders? If so, what is the paw thickness of the third founder?

Response: With respect to the IFN α levels mentioned in the heading, this was a mistake and has since been removed. IFN α had been analyzed from too few of the mice and thus was not appropriate to include. The IFN α level comparisons between the N154S and WT mice are displayed in Fig. 4F.

With regard to the 3rd founder's paw thickness, these looked to be somewhat less than the 1504 line, however, this was not significant. We included the 1504 line merely as an example as these other lines were not part of the main experiments / endpoints that were utilized for this study. The phenotypic findings were consistent between all independent founders that expressed the mutant hSTING transgene, despite the fact that each founder was the result of a random chromosomal integration of the Vav1-hSTING-N154S transgene.

10. Figure 7&8: More mice hSTING-N154S *Ifnar1*^{-/-} mice should be analyzed in the rescue experiments.

Response: With the extra time allowed us by the Editor, we were able to complete further experiments that the reviewer requested for our manuscript. We have thus significantly increased the N-numbers as the reviewer suggested – as follows (see underline text below):

“Figure 8. The hSTING-N154S vasculopathy is prevented by the absence of *ifnar1*. (A-C) Representative photographs of male age-matched WT (left), hSTING-N154S (centre), and hSTING-N154S/*ifnar1*-KO (right) mice showing differences in body size (A), ear pathology (B) and paw thickness (C). Arrows and arrowheads indicate differences in the hind-paws and ears, respectively, of hSTING-N154S mice either in the presence (red) or absence (black) of *ifnar1*. Scatter-plots showing the differences in body weight (D), paw thickness (E) and spleen weight (F) amongst WT (open circles), hSTING-N154S (closed circles) and *ifnar1* (hSTING-N154S/*ifnar1*-KO (closed squares) male and female mice. Each symbol represents an individual mouse (D-F) and horizontal lines represent the mean \pm s.e.m. Spleen weight (F) was normalized to body weight (mg/g of body weight). Statistical significance between data sets (D-F) was assessed by one-way ANOVA followed by Tukey's multiple comparisons post-hoc test between all groups. Significant differences between WT (M/F, n=10) and hSTING-N154S (M, n=7; F, n=8) or between hSTING-N154S and hSTING-N154S/*ifnar1*-KO (M, n=14; F, n=16) mice are denoted by *** $P < 0.001$.”

For Figure 9, formerly figure 8, the Luminex assay numbers have been significantly increased for the N154S/*ifnar1*-KO mice. "Each symbol represents an individual mouse and horizontal lines represent the mean \pm s.e.m., n=3 for WT and hSTING-N154S; n=8 for N154S/*ifnar1*-KO.

11. Figure S2: Histology of the respective controls should be shown to allow a broad range of readers to immediately identify aberrant effects in hSTING-N154S mice.

Response: We do not think this is necessary, since the very small infiltrates in tissue settings that are otherwise normal are clearly indicated by the arrows.

12. Figure 2: What is the age of the mice? Is there inflammation in lungs of aged mice? Do the rare inflammatory foci expand?

Response: This question appears to refer to Fig. S2, not Fig. 2? This is an image of an aged mouse. There was no significant inflammation or fibrosis in the lungs of our N154S mice and thus why we hypothesized that lung parenchymal cell expression of hSTING-N154S would be required for pulmonary inflammation. As mentioned in the figure legend: “Small rare foci of leukocytic infiltrates in the lungs of hSTING-N154S mice (arrows), in a perivascular and peribronchiolar distribution. A focus of fibrosis (B) is occasionally associated with the infiltrate.”

Aging the mice, as suggested by the reviewer, is limited owing to the development of severe paw inflammation and swelling as well as by the development of acral necrosis; thus, aging the mice out to 6 months or 1 year is not possible.

13. Figure 4: Can the authors provide data on the cellularity of the spleen, thymus and lymph nodes? Can the authors provide absolute numbers, as these are more informative and would probably reveal a more drastic effect of transgene expression on the lymphoid compartment? Why have B cells and myeloid cells not been analyzed in these organs? The legend says group size >10, but STING KO is n=4, please correct. Figure 4 D&E: Please only provide the total number of mice per group that has been analyzed in five experiments.

Response: We have shown results examining CD19⁺ and CD11b⁺ cells. The lymphoid cell perturbations caused by mutant mSting have already been thoroughly characterized. The results we present regarding lymphocytes and other cell populations are consistent with what the other groups have found regarding effects of activated STING. A comprehensive functional and flow cytometry follow-up study is planned that will focus, not only on lymphocytes and other hematopoietic cells, but also on responses to antigenic challenge, APC function and immunoglobulin levels and isotypes; especially since autoimmunity is in evidence in the mutant hSTING model.

(The possible reasons for the lack of lung inflammation in our mutant hSTING transgenic mice is addressed in the discussion.)

The group size regarding the STING-KO mice (in bold) and Figure 5 (formerly Figure 4 D&E) has been changed as suggested. Please see new Figure 5 legend.

“Figure 5. T cell lymphopenia and type I interferon levels in hSTING-N154S mice.

(A-C) While CD4⁺ T cell numbers were moderately reduced in the thymi of hSTING-N154S mice (A), there were marked reductions in the number of CD4⁺ and CD8⁺ cells in the spleens (B) and lymph nodes (C) when compared to WT mice. There were no differences in these T cell populations when WT and STING-KO mice were compared. One-way ANOVA with Tukey’s multiple comparisons post-tests were used to analyse group differences. WT and hSTING-N154S, n ≥10; STING-KO, n=4. Horizontal lines represent the mean ± s.e.m. with significant differences denoted as *P < 0.05 or ***P < 0.001 vs WT. (D) Using quantitative

RT-PCR analysis of splenic tissues, we observed that IFN- β transcripts were modestly increased in the *hSTING*-N154S mice (n=7) relative to those in WT littermates (n = 7). (E) 13-plex Luminex assay of serum showed that mIFN- β levels were elevated in the sera of 8 of 13 *hSTING*-N154S mice (n=13) compared to 4 of 10 WT littermates (n=10). (F) Compared to WT littermates, there was also a significant increase in mIFN- α levels as detected via ELISA in the sera of *hSTING*-N154S mice (n=12) (LumiKine Xpress mIFN- α ELISA kit). Horizontal lines represent the mean \pm s.e.m. serum concentration (pg/mL) of murine IFN- β or IFN- α . Data are pooled from 5 independent experiments (n=1-5 for each group). Unpaired t-test was carried out between *hSTING*-N154S and WT groups where * $P < 0.05$.”

Data on the cellularity of the thymus, spleen and lymph nodes was not recorded.

14. Figure 5: The legend says n=9 but number of symbols does not match. The legend also refers to Figure S2 but the linked dataset is represented in Figure S3. Please also harmonize the group sizes in Figures 5 and S3, as they appear to show analysis of the same mice. Also indicate how many independent experiments have been carried out.

Response: Figure 5 is now Figure 6. Values, symbols etc. have been clarified as suggested. Analysis of the Luminex results were from the same mice. We included the supplemental to ensure transparency of the results of the 31-plex assay. As is now shown in the manuscript:

Figure 6. Increased cytokine levels in *hSTING*-N154S mice.

Serum cytokine levels in *hSTING*-N154S mice (n=13) and WT littermate controls (n=12) were measured by 31-plex murine cytokine array. Each symbol represents an individual mouse and horizontal lines represent the mean \pm s.e.m. of serum concentrations for each cytokine (pg/mL). Data was pooled from 5 independent experiments (n=1-5 for each group). Unpaired t-test was carried out between *hSTING*-N154S and WT groups where * $P < 0.05$, ** $P < 0.01$, *** $P < 0.001$ and **** $P < 0.0001$. See Fig S3 for the remainder of the Luminex results from these mice.

Figure S3. Systemic cytokine levels in *hSTING*-N154S mice. Serum cytokine levels in *hSTING*-N154S mice (closed circles, n=13) and WT littermate controls (open circles, n=12) were measured by a Luminex 31-plex murine cytokine array. Each symbol represents an individual mouse and horizontal lines represent the mean \pm s.e.m. serum concentration for each cytokine (pg/mL). Data was pooled from 5 independent experiments (n=1-5 for each group). Unpaired t-test was carried out between *hSTING*-N154S and WT groups where * $P < 0.05$.

15. I would suggest to combine all data on lymphopenia in one section and to not revisit this topic after discussion of different topics. My suggestion would be to reorganize the manuscript into fewer defined sections addressing the macroscopic phenotype, hematopoietic parameters, cytokine expression, autoimmunity and the role of *Ifnar* signaling in the pathology.

Response: We appreciate the suggestion, but we prefer to keep manuscript organization as is.

16. Figure 6B - left: What is the CD19intermediate/Ly6G-positive population? How were the gating and/or compensation for the staining performed?

Response: The gates for CD19 and Ly6G were set based on known cell populations which express these markers. Neutrophils are characterized as Ly6G+ and B cells are characterized as CD19+. We have acknowledged this in figure legend 7: “Regarding the intermediate CD19+/Ly6G staining (B), there are no known leukocytes that co-express CD19 and Ly6G, and seeing that this population does not exist in the peripheral blood dot plots, this likely represents uncharacterized cells within the abundant lung parenchyma/pneumocyte populations that are shown in the scatter plots.

17. Figure S4: Based on which criteria has the gate been set? What is the CD11b high F4/80 low population that is only present in the WT but almost absent in the hSTING-N154S lungs?

Response: The CD11b+ /F480 med/low cells could be CD11b+ dendritic cells or monocytes... and possibly pulmonary interstitial macrophages. It could also be the result of the mutant hSTING-N154S inducing differentiation of F4/80 low CD11b+ high monocytes (as seen in the WT) into pro-inflammatory CD11b+ intermediate F4/80 high macrophages (as seen in the hSTING mice).

18: Figure S5: At which age have the mice been analyzed? Can the authors provide the full dataset of the immune-fluorescence pattern of the ANAs and on the auto-antigens that they tested, including titers for the reactivity against the auto-antigens? Are there ANAs to human STING in these mice? Can the authors provide any statistics on whether the ANA frequency really differs between the groups?

Response: The full dataset has been provided as requested (Supplementary figure 6). Statistics on ANAs and titers would be difficult to carry out since it would have to encompass not only ANA titers, but also ANA pattern type, and specific auto-antibody reactivities. We believe that the differences are best captured and visualized by including the raw data in the Supplemental Table that we now provide, that includes sex, ANA staining pattern, ANA titer, and autoantibody reactivities across a wide panel of substrates.

As to the ages of the male and female mice used for the serum samples for ANA testing, whenever an animal was euthanized due to disease or for other reasons, a serum sample was collected. This involved serums from experimental and WT mice between 15 and 25 weeks of age.

We have not determined whether any autoantibodies against the human STING were present in the mice. The hSTING protein would be expressed in the hematopoietic/lymphoid compartment, beginning as soon as cells of the innate and adaptive immune systems were developing. Hence, one would obviously predict that tolerance would develop, preventing an immune response against the transgene-derived protein. For this reason, it seems extremely unlikely that there would be an anti-human STING T or B cell response.

As requested, a full data set is now provided, with the following table legend:

“Autoantibodies to intracellular inflammatory myopathy (IM) and other connective tissue disease auto antigens as detected by addressable laser bead immunoassays showed increased reactivity to a variety of IM targets but the highest titers were directed to Jo-1 (histidyl tRNA synthetase), PL-7 (threonyl tRNA synthetase), Signal Recognition Particle (SRP) and U1-ribonucleoprotein (U1-RNP). Numerical values are expressed as median fluorescence index (MFI).”

19. Figure 7: Are littermate controls shown? Do all mice have the same gender? In 7D, is the difference in body weight between WT and hSTING-N153S not statistically significant? Lines above the groups that have been compared could alternatively label comparisons, which might provide a better instant understanding of the graphs.

Response: Though mice were sacrificed at various age points, all comparisons made between the different phenotypes were done using sex and age-matched littermate controls (e.g. images of body size, pathology, Luminex, etc. was that of age/sex-matched littermates sacrificed at the same time point).

With regards to “is the difference in body weight between WT and hSTING-154S not statistically significant?”, the main point here was to show that crossing the N154S onto an *Ifnar1*-KO background alleviated disease. We have added an asterisk to show that the WT is indeed different from the hSTING-N154S group. We have also increased the n values.

June 5, 2019

RE: Life Science Alliance Manuscript #LSA-2018-00215-TR

Dr. Gary R Martin
University of Calgary
Department of Biochemistry and Molecular Biology
3A25 HRIC
3280 Hospital Drive NW
Calgary, AB T2N 4Z6
Canada

Dear Dr. Martin,

Thank you for submitting your revised manuscript entitled "An hSTING mutation (N154S) in hematopoietic cells produces an Ifnar1-dependent vasculopathy in mice". As you will see, the reviewers appreciate the introduced changes, though reviewer #1 would have expected better insight into the mode of cell death occurring. We have discussed your work in light of the reviewer comments again and while we agree with the reviewer that such insight would be useful to the field, we decided that a second round of experimental revision is not needed. We would thus be happy to publish your paper in Life Science Alliance pending final revisions necessary to meet our formatting guidelines:

- please mention n of mice for Fig 1A/B
- please add a callout to Fig 4D and Fig S2B in the manuscript text
- please add scale bars in your figures

A. FINAL FILES:

-- High-resolution figure, supplementary figure and video files uploaded as individual files: See our detailed guidelines for preparing your production-ready images, <http://www.life-science->

alliance.org/authors

B. MANUSCRIPT ORGANIZATION AND FORMATTING:

Sincerely,

Reviewer #1 (Comments to the Authors (Required)):

I think the work has been significantly improved in revision. I am a bit surprised that the authors have not conclusively addressed my point #3, where I ask for characterization of the mode of cell death occurring in the SAVI mice. This is important, since the STING-death pathway may well be important for the pathology.

Reviewer #3 (Comments to the Authors (Required)):

The authors have extensively worked on the manuscript and addressed all important points carefully. In my opinion clarity and impact of the manuscript have been significantly improved and it will be of interest for a broad range of readers. It has reached the level necessary for publication in Life Science Alliance, which i recommend.

June 12, 2019

RE: Life Science Alliance Manuscript #LSA-2018-00215-TRR

Dr. Gary R Martin
University of Calgary
Department of Biochemistry and Molecular Biology
3A25 HRIC
3280 Hospital Drive NW
Calgary, AB T2N 4Z6
Canada

Dear Dr. Martin,

Thank you for submitting your Research Article entitled "A constitutively-active human STING mutant produces an Ifnar1-dependent vasculopathy". It is a pleasure to let you know that your manuscript is now accepted for publication in Life Science Alliance. Congratulations on this interesting work.

DISTRIBUTION OF MATERIALS:

Again, congratulations on a very nice paper. I hope you found the review process to be constructive and are pleased with how the manuscript was handled editorially. We look forward to future exciting

submissions from your lab.

Sincerely,
